# A 3D-model inversion of methyl chloroform to constrain the atmospheric oxidative capacity

Stijn Naus[1], Stephen A. Montzka[2], Prabir K. Patra[3,4], and Maarten C. Krol[1,5]

[1]Meteorology and Air Quality, Wageningen University and Research, the Netherlands
[2]NOAA Global Monitoring Laboratory, Boulder, CO, USA
[3]Research Institute for Global Change, JAMSTEC, Yokohama, Japan
[4]Center for Environmental Remote Sensing, Chiba University, Chiba, Japan
[5]Institute for Marine and Atmospheric Research, Utrecht University, the Netherlands

**Correspondence:** Stijn Naus (s.naus@sron.nl)

**Abstract.** Variations in the atmospheric oxidative capacity, largely determined by variations in the hydroxyl radical (OH), form a key uncertainty in many greenhouse and other pollutant budgets, such as that of methane ($CH_4$). Methyl chloroform (MCF) is an often-adopted tracer to indirectly put observational constraints on large-scale variations in OH. We investigated the budget of MCF in a 4DVAR inversion using the atmospheric transport model TM5, for the period 1998-2018, with the objective to derive information on large-scale, interannual variations in atmospheric OH concentrations.

While our main inversion did not fully converge, we did derive interannual variations in the global oxidation of MCF that bring simulated mole fractions of MCF within 1-2% of the assimilated observations from the NOAA-GMD surface network at most sites. Additionally, the posterior simulations better reproduce aircraft observations used for independent validation, compared to the prior simulations. The derived OH variations showed robustness with respect to the prior MCF emissions and the prior OH distribution over the 1998 to 2008 period. Although we find a rapid 8% increase in global mean OH concentrations between 2010 and 2012 that quickly declines afterwards, the derived interannual variations were typically small (<3%/year), with no significant longterm trend in global mean OH concentrations.

The inverse system found strong adjustments to the latitudinal distribution of OH, relative to widely used prior distributions, with systematic increases in tropical, and decreases in extra-tropical OH concentrations (both up to 30%). These spatial adjustments were driven by intrahemispheric biases in simulated MCF mole fractions, which have not been identified in previous studies. Given the large amplitude of these adjustments, which exceeds spread between literature estimates, and a residual bias in the MCF intrahemispheric gradients, we suggest a reversal in the extratropical ocean sink of MCF in response to declining atmospheric MCF abundance (as hypothesized in Wennberg et al. (2004)). This ocean source provides a more realistic explanation for the biases, possibly complementary to adjustments in the OH distribution.

We identified significant added value in the use of a 3D transport model, since it implicitly accounts for variable transport and optimizes the observed spatial gradients of MCF, which is not possible in simpler models. However, we also found a trade-off in computational expense and convergence problems. Despite these convergence problems, the derived OH variations do result in an improved match with MCF observations relative to an interannually repeating prior for OH. Therefore, we consider that variations in OH derived from MCF inversions with 3D models can add value to budget studies of longlived gases like $CH_4$.

## 1 Introduction

The hydroxyl radical (OH) is the main atmospheric oxidant and plays an integral role in atmospheric chemistry. OH is involved in the removal of a wide variety of toxic pollutants (carbon monoxide, nitrogen oxides), greenhouse gases (e.g. methane ($CH_4$) and HFC's) and gases that contribute to stratospheric ozone depletion (various HCFC's). Therefore, in order to better understand and constrain these different pollutant budgets, robust constraints on OH are required.

Due to its high reactivity, OH has a short atmospheric lifetime of seconds and is present at low abundances in the atmosphere. This makes direct measurements of the OH concentration (denoted as [OH]) difficult, and extrapolation of these measurements to global scales near impossible. Bottom-up modeling of OH in full-chemistry models is a useful tool to better understand the OH budget. However, this modeling relies on accurate understanding of the complex chemistry involved and of the emissions of many species, and it lacks observational constraints. Variations in atmospheric oxidation can also be estimated indirectly from observed variations in the growth rate of a tracer removed by OH. This "proxy"-method can provide independent observational constraints on OH on large spatio-temporal scales, against which the understanding implemented in full-chemistry models can be tested.

Different species have been proposed and used to constrain OH, with a particular focus on global mean OH concentrations ($[OH]_{GM}$) (Dentener et al., 2003; Krol et al., 2008; Liang et al., 2017). The most widely adopted of these is methyl chloroform (MCF). MCF was used as a solvent up to the early 1990s, after which its production was phased out in the Montreal Protocol and its subsequent amendments (McCulloch and Midgley, 2001; Rigby et al., 2013). MCF is predominantly removed through oxidation by OH, with secondary sinks by stratospheric photolysis and oceanic hydrolysis, resulting in an atmospheric lifetime of 5-6 years (Chipperfield and Liang, 2013). As MCF is emitted purely anthropogenically and its production is thought to be well-constrained, MCF was identified early on as a potential tracer for OH (Lovelock, 1977; Prinn et al., 1987). This potential improved after the production phase-out and the rapid emission decline, as the MCF budget became dominated by its OH sink. The combination of a period with well-constrained production, followed by a period with a small role of its emissions, makes MCF a good tracer for OH.

Study of the MCF budget has provided important insights into the atmospheric distribution of OH and temporal variations therein. For example, observations of MCF were used to quantify a likely upper limit to interannual variations in $[OH]_{GM}$ of 2-3% (Montzka et al., 2011), which is in line with full-chemistry models. On the other hand, whereas chemistry models typically find higher OH concentrations in the Northern than in the Southern Hemisphere, a modeling study found MCF observations to be most consistent with interhemispheric parity (Patra et al., 2014).

Use of MCF to constrain $[OH]_{GM}$ and the OH distribution is not without challenges, however. Through improvements in measurement techniques, measurement quality has mostly kept pace with the atmospheric decline of MCF, but artifact-free sampling has become more difficult. Additionally, a decline of the atmospheric burden implies an increasingly important role of any small, persistent emissions in the MCF budget. As a further complication, atmospheric decline of MCF has been suggested to result in a reversal of the small but significant ocean sink of MCF at high latitudes (Wennberg et al., 2004). Finally, the

surface measurement networks that monitor MCF are relatively sparse and thus they provide limited capacity to distinguish one budget term from another.

These limitations were emphasized in two recent studies that constrained interannual variations of $[OH]_{GM}$ in a two-/three-box model of MCF and $CH_4$ (Turner et al., 2017; Rigby et al., 2017). In their most likely solution, both studies suggested the possibility for an important contribution from OH to $CH_4$ growth rate variations. However, the looseness of derived constraints on OH variations also allowed for a solution with no variations in OH. In an extension of these studies, we investigated in previous work how the use of a relatively simple box model, rather than a more sophisticated 3D transport model, could have

affected these conclusions (Naus et al., 2019). We found that large changes in the MCF budget over time (i.e. the sudden drop in its emissions) resulted in significant changes in, for example, interhemispheric transport of MCF and the stratospheric MCF sink. However, accounting for these changes in our two-box model did not alter the conclusion that MCF-derived constraints on multi-annual variations of OH are too uncertain to determine the exact contribution of OH to the relatively small but important $CH_4$ growth rate variations.

In this study, we present an inversion of MCF in the 3D chemistry-transport model TM5 aimed at constraining $[OH]_{GM}$ and the OH distribution. The advantage of approaching the problem in a 3D transport model, instead of in a box model, is two-fold. Firstly, by explicitly resolving transport, we avoid the transport biases that hamper simple box models. Secondly, we can fully exploit the available observations and the gradients between surface sites. For example, since MCF is predominantly removed in the tropics, the latitudinal distribution of MCF has had a tropical minimum since 1998 (Spivakovsky et al., 2000;

Montzka et al., 2000). There is potential for this type of information to provide constraints on OH that has not been exploited by box-model approaches.

    The last comprehensive 3D model OH-inversion of MCF investigated the 1980-2000 period (Bousquet et al., 2005). Here, we instead cover the 1998-2018 period. An important difference between these two periods concerns the role of emissions. The 1980-2000 period includes a period with high MCF emissions, as well as the subsequent strong decrease in emissions

from 1990 onward, which complicated the interpretation of MCF observations. In contrast, the situation over 1998-2018 was simpler, with a near-constant exponential decline of atmospheric MCF and a small role for its emissions.

    The objective of this study is to investigate information on large-scale variations in OH concentrations contained in measurements of the most promising tracer identified to date, MCF, during the period that follows its drop in emissions, using the most comprehensive tools available to us, in the form of a state-of-the-art inverse system built around a 3D transport model.

## 2 Methods

### 2.1 Inverse method

#### 2.1.1 TM5-4DVAR

To simulate atmospheric transport, we used the transport model TM5 (Krol et al., 2005). We operated TM5 on a 6° by 4° horizontal resolution, with 25 vertical layers. Transport in TM5 was driven by offline meteorological fields from the ERA-Interim reanalysis from the European Centre for Medium Range Weather Forecasts (ECMWF) (Dee et al., 2011).

For optimization of OH concentrations and other parameters, we used the inversion framework TM5-4DVAR (Meirink et al., 2008). TM5-4DVAR is an inverse system based on the variational optimization technique 4DVAR. TM5-4DVAR has been applied successfully in previous studies to a variety of problems, such as to constrain the global $CH_4$ budget (Bergamaschi et al., 2009) and to investigate the Indonesian wildfires (Nechita-Banda et al., 2018). The objective of our set-up of the TM5-4DVAR inverse system is to find the optimal configuration of MCF emissions and OH variations that best reproduce atmospheric observations of MCF. Formally, this objective is quantified as minimization of the cost function $J$ (Equation 1).

$$J(\boldsymbol{x}) = \frac{1}{2}(\boldsymbol{x} - \boldsymbol{x}_{prior})^T \mathbf{B}^{-1}(\boldsymbol{x} - \boldsymbol{x}_{prior}) + \frac{1}{2}(\mathbf{H}\boldsymbol{x} - \boldsymbol{y})^T \mathbf{R}^{-1}(\mathbf{H}\boldsymbol{x} - \boldsymbol{y}). \tag{1}$$

$J$ is a function of the state $\boldsymbol{x}$, which contains all the parameters to be optimized, such as OH concentrations. The cost consists of two terms. First is the deviation from the first guess $x_{prior}$, weighted by the prior error covariance matrix $\mathbf{B}$. Second is the difference between simulated MCF mole fractions, calculated in the forward version of TM5 (denoted as $\mathbf{H}$), and the real-world observations $\boldsymbol{y}$, weighted by the observational covariance matrix $\mathbf{R}$. Additionally, in the 4DVAR optimization, the gradient of the cost function $\nabla J$ is calculated and used (Equation 2).

$$\nabla J(\boldsymbol{x}) = \mathbf{B}^{-1}(\boldsymbol{x} - \boldsymbol{x}_{pri}) + \mathbf{H^T}\mathbf{R}^{-1}(\mathbf{H}\boldsymbol{x} - \boldsymbol{y}), \tag{2}$$

with $\mathbf{H^T}$ the adjoint of the forward model $\mathbf{H}$. The adjoint of TM5 is extensively described in Meirink et al. (2008) and Krol et al. (2008). Since OH chemistry is non-linear and since we optimized emissions non-linearly (see Section 2.1.2), $\mathbf{H^T}$ is actually the adjoint of the forward tangent linear model. The derivation of the adjoint OH chemistry is described in Supplement S6.

#### 2.1.2 Inversion set-up

In this section we discuss the set-up of the three inversions we performed. First we describe the set-up of the standard inversion (hereafter referred to as REF); next we describe the corresponding $\mathbf{B}$ matrix; finally we describe the two variations of the REF inversion which we performed. Note that the $\mathbf{R}$ matrix is discussed in Section 2.2.

In the REF inversion, we used the MCF source and sink fields from the TransCom-CH4 project (Patra et al., 2011). Loss fields for OH, stratospheric photolysis and ocean uptake, as well as initial fields of MCF, are described in the TransCom-

CH$_4$ protocol. Briefly, the OH fields are a combination of tropospheric OH fields from Spivakovsky et al. (2000), scaled by a factor 0.92, and stratospheric OH fields derived with the 2D MPIC chemistry model (Brühl and Crutzen, 1993). The ocean flux is a first-order sink proportional to MCF mole fractions in the lowest model layer and a spatially variable uptake rate, which maximizes in the tropics. Stratospheric photolysis fields were generated with simulations of the ACTM model (Patra et al., 2009). TransCom MCF emissions were available only up to 2006. In subsequent years, we repeated the 2006 spatial distribution.

We pre-optimized prior MCF emissions to fit global mean MCF mole fractions, as derived from the NOAA surface network, assuming no variations in OH. The procedure to obtain global mean MCF mole fractions from the NOAA surface network is described in Supplement S1 of Naus et al. (2019). We subsequently removed year-to-year variations in the resulting fitted MCF emissions with a three-year moving average. We motivate this choice of pre-optimization as follows. Firstly, the absence of strong prior constraints on MCF emissions over part of the inversion period necessitates some arbitrary choice for the emission prior. Secondly, we find that, if the prior simulation diverges too strongly from real-world observations (for example because MCF emissions are assumed to level out at a too-high value in later years), the computational cost of the inversions is greatly increased. Thirdly, while we technically made double-use of MCF observations through the pre-optimization, interannual variations and spatial gradients in MCF observations remain unused, and these are important quantities in the 3D model inversion.

We included scaling factors for both OH concentrations and MCF emissions in the state vector for optimization. Emissions were optimized monthly in each grid box with a horizontal correlation length of 500 kilometres and a temporal correlation of 9.5 months. We adopted a grid-box error in emissions of 50% up to 2005 that increases with 15% per year, up to 200% in 2015. This increase ensures that when prior emissions become low , the absolute error on emissions remains large (around 4 Gg/year), which reflects the lack of prior constraints on MCF emissions in later years. In the construction of the **B**-matrix, OH concentration fields were optimized monthly, in 45 latitudinal bands (i.e. on our native latitudinal resolution), with an error per band of 10%. We imposed high correlations between the latitudinal bands of 0.8 (corresponding to a correlation length of 1500-2000 km), and a temporal correlation length of 12 months. Since there are fewer state elements for OH than for emissions, we increased the cost of adjusting OH relative to that of adjusting emissions by an additional factor 5, similar to e.g. Maasakkers et al. (2019). This adjustment ensures that the cost of adjusting total emissions or [OH]$_{\mathrm{GM}}$ by 1-$\sigma$ of the global uncertainty is comparable.

Both OH concentrations and emissions were optimized with non-linear scaling factors, which preclude negative emissions and OH concentrations. This method was introduced in Bergamaschi et al. (2009) and we describe it more extensively in Supplement S5. Due to the non-linear nature of this inverse problem, we use the M1QN3 solver (Gilbert and Lemaréchal, 1989). The downside of this choice is that we cannot retrieve the posterior covariance matrix. We instead tested the robustness of derived solutions with respect to the OH and the emission distribution, two very likely sources of uncertainty, in two additional inversions.

In our second inversion, referred to as POP, we redistributed the same annual total MCF emissions as in the REF inversion proportional to population density (as retrieved from CIESIN, Columbia University (2018)). In the third inversion (referred

to as TM5OH), we used the same emissions as in the REF inversion, but adopted a tropospheric OH distribution based on a simulation of the year 2006 performed with the full-chemistry version of TM5 (Huijnen et al., 2010), combined with the same stratospheric distribution as in the REF inversion. Differences between the two OG distributions are typically 10-15%, depending on the latitude. The TM5OH distribution has relatively higher OH concentrations in the Northern hemisphere (see Figure S8).

On a final note, while we optimized scaling factors for OH concentrations per latitudinal band, we prefer to discuss the change in global mean oxidation in further sections, rather than the change in $[OH]_{GM}$. We quantified the change in oxidation as the atmospheric mass-weighted average of $k(T) \cdot [OH]$, with $k(T)$ the temperature-dependent reaction rate between OH and MCF (Burkholder et al., 2015). This is necessary, because we allowed for adjustments in the latitudinal distribution of OH. Since latitude and temperature are strongly correlated, a latitudinal redistribution of OH that conserves global total oxidation often implies a change in $[OH]_{GM}$. In that case, we consider the conservation of global total oxidation the relevant quantity, rather than a change in $[OH]_{GM}$ (similar to the recommendation in Lawrence et al. (2001)). We calculate the variations in oxidation relative to the prior, so that for example interannual variations of temperature will not affect these variations in oxidation, since temperature variations remain the same between prior and posterior simulations. Where relevant, we make note of this distinction. We also present results for the latitudinal adjustments in OH concentrations.

## 2.2 Observations

### 2.2.1 Surface observations

We used MCF observations from the surface network of the National Oceanic and Atmospheric Administration (NOAA) Global Monitoring Laboratory (GML) as the only observational constraints in the inversion. The network consists of a core set of 7 surface sites that have monitored MCF since 1992, and additional sites have been added since: data from a total of 12 sites are available since 1998. The sites we used, including site abbreviations, are described in Table S2. At each site, paired flask samples are collected at weekly to monthly frequency, following a sampling protocol that typically favors sampling under meteorological conditions that correspond to clean background air. Flask samples are then collected and measured on one central measurement system against the NOAA calibration scale for MCF. The measurement uncertainties we used are those reported by NOAA, which are based on the difference between the mole fractions measured for each flask in a flask pair. Up to 2018, short-term measurement repeatability remained consistently around 0.5% of the measured mole fraction. On top of the measurement error, we also included a model representativeness error for each observation. This error is calculated as an absolute average over the mole fraction gradients between the model grid cell that contains an observation and horizontally and vertically adjacent grid cells (Bergamaschi et al., 2005). Typically, the model error was up to 1% at Northern midlatitude sites (e.g. LEF), and as low as 0.1% at e.g. SPO or ALT. The addition of these two error sources, with no correlations in between, constitute the **R** matrix (see Section 2.1).

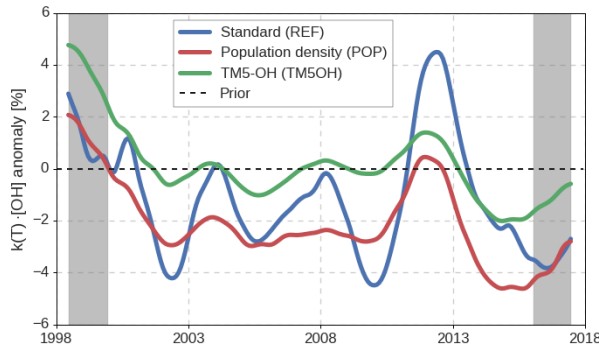

**Figure 1.** Monthly variations in global mean $k(T)\cdot[OH]$, derived in the three different inversion set-ups. Variations in $k(T)\cdot[OH]$ are calculated relative to the prior simulation, thus removing the influence of interannual variations in temperature. Gray bars indicate a spin-up and spin-down period of 1.5 years.

### 2.2.2 Aircraft campaigns

For validation of the inversion results, we used two sets of aircraft campaigns: the HIAPER Pole-to-Pole Observations (HIPPO, 2009-2012, Wofsy et al. (2011)) and the Atmospheric Tomography Missions (ATom, 2016-2018, Wofsy et al. (2018)). The five HIPPO campaigns were flown from 85°N to 67°S over the Pacific, with vertical profiles being sampled up to 8500 meters at approximately 2.2° intervals. The horizontal extent of the ATom campaigns was wider, with global coverage over the Pacific, the Atlantic and the Arctic oceans.

While a number of different measurement systems provided measurements of MCF during HIPPO, only results from the NOAA Whole Air Sampler during H-1, -3, -4, and -5 were included in this analysis because they are most closely tied to NOAA's surface network results (i.e., in methodology, precision, and calibration consistency). ATom results from deployments A-2 and A-3 were also included in this work, as those flask samples were analyzed on the same NOAA instrument as the surface network flasks. We exclude from our analysis a subset of results from HIPPO-1, as well as all of HIPPO-2 and the ATom-1 missions, since samples from these deployments suffered from a deployment-specific measurement interference (a portion of HIPPO-1 and all of HIPPO-2) or were analyzed on a different instrument in NOAA that exhibited poorer precision (ATom-1).

## 3 Results

### 3.1 Variations in the atmospheric oxidative capacity and in MCF emissions

In Figure 1, we show the monthly anomalies in global oxidation (different from $[OH]_{GM}$: see Section 2.1.2), as derived in the three inversion set-ups. We have shown the entire twenty-year inversion period, which will include a spin-up and spin-down period of 1-2 years (indicated by the gray bars). For example, even though our initial MCF mole fraction fields are realistic,

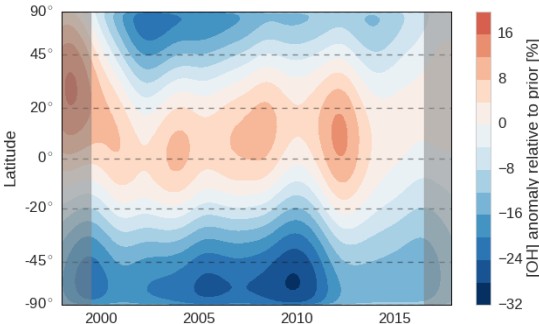

**Figure 2.** Hovmöller diagram of the spatio-temporal adjustments made to [OH] in the REF inversion set-up. The prior [OH] distribution, with respect to which changes are shown, is based on Spivakovsky et al. (2000), scaled down by a factor 0.92. Gray bars indicate a spin-up and spin-down period of 1.5 years.

the strong positive oxidation anomaly in 1998 might be linked to errors in the initial field. The corresponding variations in the
tropospheric lifetimes of MCF and of $CH_4$ are described in Supplement S1.

Interannual variations in global, annual mean oxidation that we derive are typically small ($\sim 2\%$). In this, there is consistency between the different inversion set-ups. Variability in global oxidation derived in the POP and the TM5OH inversions are strongly positively correlated ($R^2 = 0.92$), while the correlation with the REF inversion is weaker ($R^2 = 0.51$). It is counter-intuitive that the REF inversion deviates from the other two, as it shares its emission distribution with the TM5OH inversion,
and its OH distribution with the POP inversion. In contrast, the POP and TM5OH inversions share neither. Therefore, we consider a large part of the differences attributable to convergence problems, caused by the size of the inverse problem, which we discuss further in Section 3.5. We consider the REF solution best converged, since its solution results in the best match with observations (see Section 3.2), and because the REF solution is most consistent with a set of ten-year inversions (covering 1998 to 2008) that converged to a higher degree (see Supplement S5).

As outlined in Section 2, we optimized OH concentrations in 45 latitude bands of $4°$ each. Figure 2 shows the resulting adjustments per latitude band through time, for the REF inversion. Clearly, adjustments to zonal mean OH concentrations can be much larger (up to 30%) than adjustments to annual $[OH]_{GM}$ (up to 5%). Moreover, there is a strong systematic tendency to increase tropical OH concentrations, and decrease extra-tropical OH concentrations, especially in the Southern Hemisphere. This tendency was observed in each of the three inversion set-ups, i.e. also when a different OH distribution was used. We
further investigate this tendency in Sections 3.2 and 3.4. The prior and posterior latitudinal OH distributions are compared to a range of literature estimates in Figure S8.

The MCF emissions that result from the three inversions are shown in Figure 3. In general, emissions are increased relative to the prior, especially in later years. The REF inversion shows the largest adjustments and pronounced interannual variations, with even an increase in some years superimposed on the downward trend. Firstly, we note that the small emission totals in
later years of around 2 Gg/yr, with interannual variations of 0.2 Gg/yr, would be hard to exclude based on prior knowledge

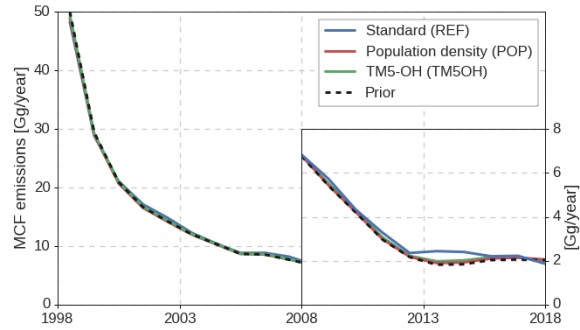

**Figure 3.** Timeseries of annual global total MCF emissions, derived in the three different inversion set-ups (solid lines) and the prior emissions (black dashed). To visualize variations in the low emissions in later years, we provide emissions in the second decade on a different y-scale (right axis).

of emissions. Secondly, the derived interannual variations in MCF emissions, relative to the prior emissions, are largely uncorrelated with derived variations in atmospheric oxidation ($r = 0.13$ in the REF inversion). A low correlation indicates that the MCF observations, to some degree, allow for independently derived $[OH]_{GM}$ and MCF emission variations. However, the contribution of the emissions to the MCF budget increases over the inversion period. Emissions are equivalent to 12%

of the loss to OH from 2000 to 2006, but afterwards this ratio increases to up to 22% in 2017-18. Notably, we find that the emissions stabilize around 2 Gg/year in the final 5 years. This stabilization is partly driven by the assumed prior emissions (based in global mean MCF mole fractions, see Section 2.1.2), to which the inversion makes only small adjustments in later years. A growing relative contribution of emissions in the MCF budget would make derived OH variations more uncertain, especially if emission distributions had changed substantially over time. Therefore, even if high-quality measurements remain

possible, residual MCF emissions could make it increasingly difficult to derive interannual changes in $[OH]_{GM}$ from MCF. Additionally, the uncertain rate of decline of MCF emissions since 2001 (the last year with reported emissions) complicates the derivation of a trend in $[OH]_{GM}$.

## 3.2 Comparison to observations

### 3.2.1 NOAA surface network

Simulated global mean MCF mole fractions and the interhemispheric gradients in the posterior simulation match well with observed gradients (Figure 4). This illustrates the skill of the inverse framework to adjust emissions and [OH] such that large-scale gradients of MCF are very well ($< 1\%$) reproduced.

However, the inversion cannot reproduce some of the observed gradients between stations, especially gradients within hemispheres. In Figure 5, gradients between three pairs of NOAA surface sites are shown. Firstly, the interhemispheric gradient

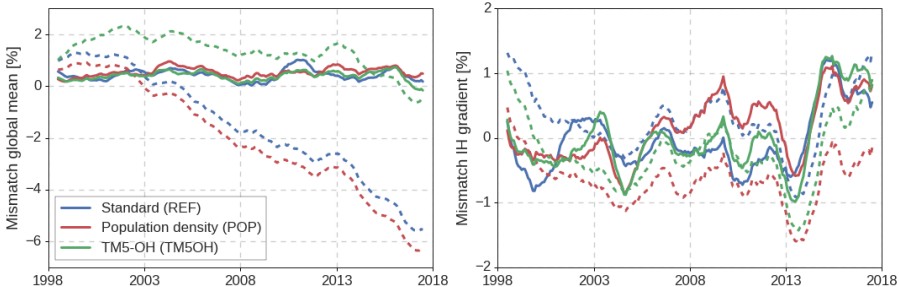

**Figure 4.** Fractional mismatch between simulated and observed MCF mole fractions in the global mean (left) and the interhemispheric gradient (right). Mismatches are given relative to global mean MCF and averaged using a twelve-month running mean. Dashed lines indicate prior mismatches, while solid lines show posterior mismatches. Global and hemispheric mean mole fractions were calculated from the NOAA surface network following the methods outlined in S1 of Naus et al. (2019).

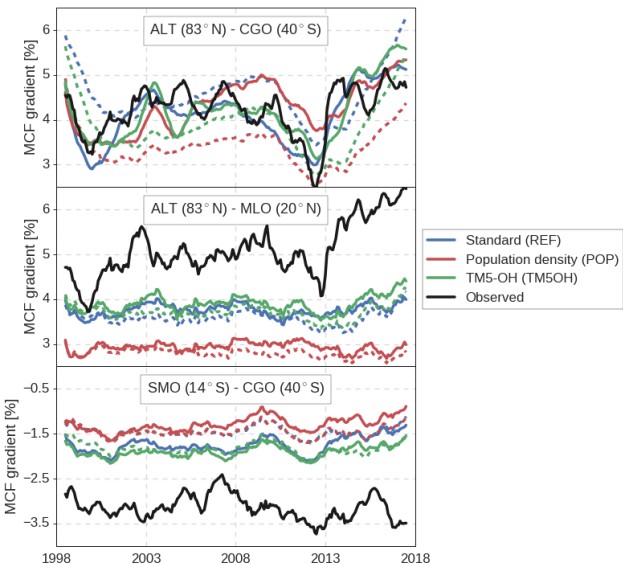

**Figure 5.** Timeseries of observed and modelled gradients between three sets of NOAA surface sites. Gradients are given relative to global mean MCF and averaged using a twelve-month running mean. Dashed lines indicate prior mismatches, while solid lines show posterior mismatches. Site latitudes are also shown and further details on assimilated surface sites are given in Table S2.

between ALT and CGO is captured well in all inversions. This interhemispheric gradient is strongly affected by emissions and therefore the inversion framework can adjust it with relative ease.

In contrast, intrahemispheric gradients are less well captured. In Figure 5 it can be seen that both the gradient within the Northern Hemisphere, between ALT and MLO, and the gradient within the Southern Hemisphere, between SMO and CGO, are underestimated. More precisely, MCF mole fractions simulated at tropical sites are systematically too high (1-2 $\sigma$, with $\sigma$ the

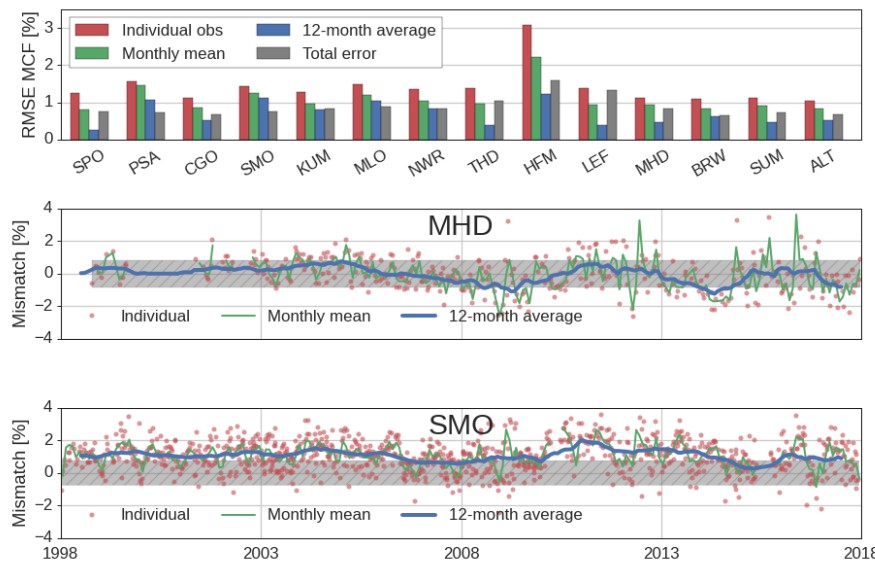

**Figure 6.** Top panel: Posterior root-mean-squared errors (RMSE) per surface site, for the REF inversion. RMSE are shown for individual observations, for monthly mean and for twelve-month running mean MCF mole fractions. Additionally, the site-dependent average total error that was used in the inversions, combined from a model and measurement error, is shown. RMSE are averages over the 1998-2018 inversion period and are given relative to the measured MCF mole fraction. Site abbreviations are explained in Table S2. Bottom two panels: Timeseries of the posterior measurement-model mismatch at MHD (top) and SMO (bottom) from individual observations, from monthly means and from twelve-month running averages, with the average total error shaded in gray.

total error), with a smaller, opposite bias at high-latitude sites (0.5-1 $\sigma$). The large latitudinal adjustments of OH concentrations (see Figure 2) are an attempt by the inversion to reduce this bias, but the adjustments only reduce the bias partly (Figure 5). As an example, the SMO-CGO gradient is, in the REF inversion, increased from -1.5% to -2%, i.e. an increase of 30%, which corresponds well with a 30% latitudinal adjustment in [OH]. However, the observed gradient is larger still at -3%. We investigate this residual bias in more detail in Section 3.4.

We have quantified the skill of a simulation that uses optimized OH and MCF emission distributions from the REF inversion to reproduce observed MCF mole fraction in a root-mean-squared error (RMSE) per site, averaged over the 1998-2018 period (top panel in Figure 6). We distinguish between the RMSE of individual observations (as used in the optimization, in red), the RMSE of the monthly mean values (cyan), and the RMSE of twelve-month running averages (green). This helps to disentangle the contribution of short-term versus long-term variations to the model-measurement mismatch. Also shown is the pre-defined

total observational error (gray), which we used in all inversions. The total error consists of a model error based on modelled spatial gradients and a measurement error based on short-term measurement repeatability (see Section 2.2.1). Additionally, in the lower two panels, we visualize two examples of these mismatches for MHD, a somewhat polluted site with no residual bias, and for SMO, a relatively clean site with residual bias.

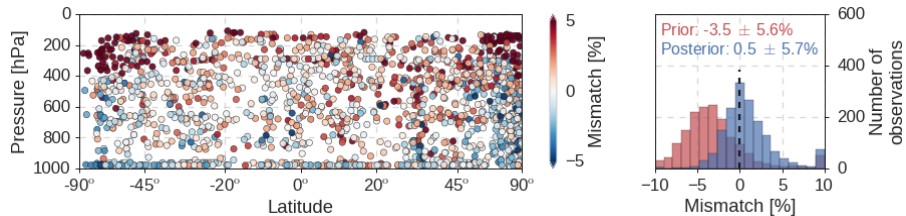

**Figure 7.** The mismatch of simulated minus observed MCF mole fractions from the HIPPO and ATom aircraft campaigns. Differences are shown relative to the observed mole fraction. We aggregated results from the HIPPO 1, 3- 5 and ATom 2 and 3 campaigns. Left: A plot of the model-measurement mismatches for each of the aircraft samples. Modelled mole fractions were sampled from the posterior REF simulation. Right: A probability-density distribution of the prior (red) and posterior (blue) mismatches in the REF inversion. Numbers denote the median mismatch and its standard deviation. Mismatches that fall outside the range on the x-axis were added to the outermost bars.

At all sites, the posterior RMSE of individual observations exceeds the total error. However, at most sites, the RMSE comes
more in line with the total error for monthly means, and especially for twelve-month running averages. This implies that, largely, the RMSE is related to short-term variations. Our inverse system, which employs relatively smooth and stiff OH and emission distributions at a coarse model resolution, has limited capability to fit short-term variations of MCF. Short-term variations in MCF are likely related to errors in the emission distribution and in small-scale transport, since OH has an integrated, slow effect on MCF mole fractions. Therefore, we do not expect these residuals to affect our OH estimate significantly. Importantly, we
point out that our initial error estimate might have been overly conservative (mostly $< 1\%$, see Figure 6), which is supported by error estimates used in previous MCF inversions (1-2% for individual observations (Bousquet et al., 2005); 5% for monthly, hemispheric averages (Turner et al., 2017; Rigby et al., 2017)). We conclude that, as long as we capture long-term variations of MCF at each site, the unresolved residuals on a sample-to-sample basis are unlikely to impact our conclusions. For the same reason, we would recommend a follow-up inversion to optimize monthly means per site, rather than individual observations.
At some sites (notably SMO, MLO and PSA), we have identified systematic biases which are of more concern, and these result in relatively large RMSE even in twelve-month running averages. We further discuss systematic offsets in Section 3.4.

### 3.2.2 HIPPO and ATom aircraft campaigns

As the inversions were driven by observations from the NOAA surface network, we used the HIPPO and the ATom aircraft campaigns as independent data sources for validation. The main added value of the aircraft over surface observations is that
the former provide snapshots of vertical gradients. When observed MCF mole fractions from all campaigns are compared to model-sampled mole fractions (Figure 7), a few features emerge.

Firstly, we find that the optimized REF simulation overestimates MCF mole fractions in the lower stratosphere at high latitudes ($> 50°$) in both hemispheres (left panel in Figure 7). This bias points to limited ability of TM5 to capture vertical gradients in the downward branch of the Brewer-Dobson circulation. A similar bias was identified in TM5 simulations of $CH_4$,
which has a stratospheric sink similar to MCF (Houweling et al., 2014).

In addition to the large overestimation at high altitudes, we also find a weak but significant positive correlation between altitude and model-measurement mismatch ($R^2$=0.12; p<0.001). Close to the surface, the average model-measurement mismatch is close to 0%, which increases to 1.5% at 10km. In other words, TM5 increasingly overestimates MCF mole fractions at higher altitudes, resulting in an underestimate of vertical gradients in TM5.

Positively, we find that the inversion improves the agreement between simulations and aircraft observations (right panel in Figure 7). While simulated mole fractions in the prior simulation are too low, in the posterior simulation MCF mole fractions compare well with observed mole fractions. The right panel in Figure 7 does show a distribution skewed towards model overestimation, which is linked to the overestimation in the lower stratosphere discussed above. While the average model-measurement difference decreases in the inversion, the standard deviation of the differences remains around 5%. This shows that vertically uniform adjustments in OH concentrations and in MCF emissions do not significantly modify simulated vertical gradients.

In conclusion, assimilation of surface observations improves agreement of our simulations with aircraft observations. However, modelled vertical gradients of MCF remain slightly smaller than those observed, although the maximum model bias is 1.5%, which is small compared to the 5% random error. Moreover, since these biases are consistent between the aircraft campaigns, we deem the impact of the biases on derived interannual and multi-annual variability of $[OH]_{GM}$ or MCF emissions small. Estimates of the total atmospheric oxidizing capacity are more likely to be affected by a systematic underestimate of vertical MCF gradients. This underestimate could be driven by an underestimate of the vertical OH gradients or by too-fast vertical mixing in TM5. While TM5 typically compares well to other transport models in terms of large-scale transport features, for example in the Age of Air experiment (Krol et al., 2018), this comparison does highlight the crucial role of aircraft campaigns in helping to identify remaining transport model biases.

### 3.3 Physical drivers of variations in atmospheric oxidation

The El Niño Southern Oscillation (ENSO), a dominant mode of natural atmospheric variability, has previously been suggested to influence interannual variations in $[OH]_{GM}$ (Prinn et al., 2001; Turner et al., 2018). ENSO affects many processes that are linked to OH, such as temperature, atmospheric moisture, lightning, wildfires and atmospheric transport.

Figure 8 shows variations in ENSO, quantified in the Multivariate ENSO Index (MEI), together with the annual, global mean $k(T)$·[OH] variations derived in the REF inversion. We find a negative correlation of -0.47 ($p = 0.05$) between the MEI and the derived $k(T)$·[OH] anomalies, if we exclude one year of spin-up and spin-down. The largest exception to the negative correlation is the positive $k(T)$·[OH] anomaly in 2012 that is not explained by a coincident variation in the MEI, which indicates that there are other controlling processes that can drive $[OH]_{GM}$ variations.

That variability in $[OH]_{GM}$ correlates with a dominant driver of atmospheric variability seems logical. However, attribution of the negative correlation to specific processes is difficult, given the large number of processes that are affected by ENSO and that in turn could affect OH. Nonetheless, we can hypothesize. For example, El Niño years (high MEI) are associated with more wildfires, resulting in higher CO emissions which could suppress OH concentrations (Zhao et al., 2020; Nguyen et al., 2020). La Niña years (low MEI) are associated with increased convection over the Pacific, increased lightning $NO_x$ production

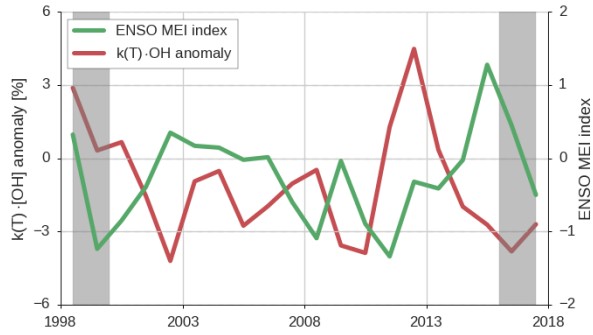

**Figure 8.** Variations in the annual mean MEI (right axis) and variations in global, annual mean $k(T)\cdot$[OH] (left axis), as derived in the REF inversion. Gray bars indicate a spin-up and spin-down period of 1.5 years.

and by extent increased OH recycling (Turner et al., 2018). The ENSO cycle additionally influences tropospheric, tropical $O_3$ abundance, although the sign of this effect can vary regionally (Oman et al., 2011).

The correlation between MEI and the $[OH]_{GM}$ variations derived in a two-box model inversion (Naus et al., 2019) is -0.01 if we do not tune the box model with a 3D transport model, and -0.34 if we do. This increase after accounting for some of the box model biases, and the even higher correlation if we move the inversion completely to a 3D transport model, shows that the correlation becomes apparent only when realistic transport of a 3D transport model is included. For example, transport variations related to ENSO have been shown to strongly affect interhemispheric differences of $CO_2$ (Francey and Frederiksen, 2016) and of $CH_4$ (Pandey et al., 2019), as well as mole fractions of MCF at SMO (Prinn et al., 1992).

### 3.4 Explaining the underestimated intrahemispheric gradients of MCF

The systematic underestimation of intrahemispheric gradients in both hemispheres deserves further elaboration, since the inversion framework has difficulties to adjust OH concentrations and emissions in such a way that intrahemispheric gradients are reproduced. MCF mole fractions are overestimated in the tropics and underestimated at high latitudes (Figure 5). To resolve these biases, the inversion introduces large adjustments in the latitudinal distribution of OH (up to 30%, see Figure 2). A set of inversions that only covered the 1998-2008 period were run to higher convergence (see Supplement S5). These inversions show that more extreme adjustments in the latitudinal OH distribution (up to 60%) better reproduce intrahemispheric gradients. However, intrahemispheric gradients were still not quite captured, and substantially higher MCF emissions were required to reproduce the gradients. The amplitude of these adjustments in the OH distribution seem physically unlikely, since the resulting OH distribution falls outside the spread of a wide range of literature estimates, as shown in Figure S8. Moreover, in none of the inversions the biases were fully resolved. Therefore, we consider here alternative, or complementary explanations for the biases, that were not explored in the inverse framework. We emphasize that the biases remain quite constant over the

twenty-year period, and while especially pronounced in the Southern Hemisphere, the biases are also present in the Northern Hemisphere. These aspects make anthropogenic emissions as a sole explanation unlikely.

One explanation for underestimated intrahemispheric gradients is that intrahemispheric mixing is too fast in TM5. Even after extensive participation of TM5 in intermodel comparisons (e.g., Krol et al., 2018), transport biases might persist. We have tested the effect of both horizontal (down to 1° x 1°) and vertical (up to 60 layers) model resolution on the simulated mole fractions (see Supplement S2). While we did find some sensitivity to resolution, the intrahemispheric biases persisted. Additionally, when we simulated $SF_6$, a chemically inert tracer commonly used for diagnosing large-scale transport (e.g., Patra et al., 1997; Denning et al., 1999), we did not find any comparable bias. Finally, we also simulated HFC-152a, a gas that is removed by OH against a shorter lifetime than MCF ($\sim$1.5 years (Ko et al., 2013)). For HFC-152a, we also found none of these biases. The results for $SF_6$ and HFC-152a are discussed more extensively in Supplement S3. We conclude that the biases seem particular to MCF and are therefore unlikely to be related to either transport or OH.

This leaves stratospheric photolysis, ocean-atmosphere exchange, another unknown MCF budget term, or a site-dependent measurement bias as potential explanations. Of these four, we consider the oceanic flux the likeliest. Wennberg et al. (2004) suggested that low MCF hydrolysis rates in cold, high-latitude oceans could result in a build-up of oceanic MCF. This reservoir would be released again when atmospheric mole fractions of MCF dropped, following production reductions demanded by the Montreal Protocol. They calculated and forecast the associated change in the ocean flux in five latitude bands for the period of 1961 to 2020 in a global ocean circulation model. We implemented their ocean flux in a forward simulation of MCF, and found that the adjusted ocean flux reduced the intrahemispheric bias by around 50% (see Supplement S4). This reduction is larger than the adjustments brought about by a 30% adjustment in tropical OH concentrations, and provides an arguably more likely scenario.

We also sampled aircraft observations in the simulation with an adjusted ocean flux. Release of MCF from high-latitude oceans would enhance vertical gradients, which combines well with the observation that TM5 underestimates aircraft-observed vertical gradients of MCF (see Section 3.2.2). However, we find that vertical gradients of MCF are only very slightly modified by the change in ocean flux, so that the signal of oceanic release of MCF in aircraft observations is too small to identify using the tools available to us.

Some new issues arise when we include the new ocean flux. Global mean MCF is overestimated, as the oceanic flux is much reduced, or even reversed. Additionally, the MCF gradient between ALT and CGO is captured less well. However, these are offsets that our inverse framework can resolve, as was found in Section 3.2.1. Therefore, we propose that a change in the oceanic flux of MCF very likely provides part of the solution to the residual intrahemispheric biases in our inversions. Further investigation of this issue, however, is beyond the scope of this study.

We note the dichotomy in the evidence for oceanic release of MCF. Positively, it emphasizes the resolving power of a sparse network of internally-consistent, low-frequency observations: we unexpectedly found a likely constraint on the oceanic uptake of MCF, in addition to the constraints on OH and emissions. Negatively, the existence of an ocean-atmosphere exchange of MCF, as opposed to a first-order sink, further increases uncertainty on any MCF-derived constraints on OH. Under the assumption that the exchange from Wennberg et al. (2004) is accurate, the impact of a different oceanic sink on derived

interannual variations of OH has been shown to be small (Prinn et al., 2005). However, the ocean-atmosphere flux is rather uncertain, since experiments that measured hydrolysis rates of MCF are scarce, especially at low ($< 20°$C) temperatures (see Supplement S4), and the hypothesis of an ocean source of MCF has not been substantiated with field measurements. Therefore, we consider that the similar spatial signature of the OH sink and the ocean uptake (both high in the tropics) will make it difficult to resolve between the two using only the surface network. Still, given the continuous decline of MCF abundance over our

inversion period, no strong interannual variations in the ocean flux are expected. Therefore, variations in OH concentrations are still the most likely driver of interannual variations in MCF mole fractions, which we consider therefore a robust feature of our inversion. Contrastingly, uncertainty in the oceanic flux does complicate the derivation of a trend in $[OH]_{GM}$ from MCF observations, as well as spatial adjustments in the OH distribution. In other words, while we deem the adjustments to the OH distribution of 30% unlikely, we cannot exclude some underestimation of tropical OH concentrations as a partial explanation

for the intrahemispheric biases.

### 3.5 The problem of convergence

Conventionally, the convergence criterion of a 4DVAR inversion is defined as a reduction in the cost function gradient. Ideally, the resulting converged state reproduces observations within the pre-defined total error. Due to pragmatic constraints, this procedure did not provide us with the desired result. Firstly, computational expense ($\sim$3 iterations in two days) limited the

385 number of iterations. Secondly, while further convergence was possible, the derived adjustments to OH concentrations became less realistic physically. Two underlying problems have been pointed out: a difficulty in defining observational and model uncertainties and difficult-to-resolve intrahemispheric biases. However, while each of the three inversions faced these two problems, they did not reach the same level of convergence. For example, an emission distribution based on population density is more diffuse, and therefore contains more degrees of freedom than the TransCom emission distribution, but the TransCom

distribution resulted in better convergence.

The importance of considering convergence when comparing the inverse results is illustrated by a set of inversions with the same set-up described in Section 2.1.2, but covering only 1998-2008. These ten-year inversions required less time per iteration and converged more consistently and in fewer iterations. The three ten-year inversions result in much more consistent interannual variations and spatial adjustments in OH concentrations, as described in more detail in Supplement S5. In turn, the

395 interannual $[OH]_{GM}$ variations and the spatial OH adjustments derived in the ten-year inversions were most similar to those derived in the twenty-year REF inversion. This suggests that the converged solutions of the three inversions are likely to be similar, but that for unclear reasons the twenty-year REF inversion managed to converge furthest, while the other two either required more iterations, or reached a false minimum.

In the end, we consider that the twenty-year REF inversion reproduces NOAA surface observations to a satisfying degree

(except for the intrahemispheric biases) and so we present the twenty-year REF inversion as our reference dataset. We do note that the amplitude of OH adjustments in the better-converged ten-year REF inversion is larger than that in the twenty-year REF inversion. For instance, interannual variability in $k(T) \cdot [OH]$ averaged over 1999-2008 (i.e. excepting the spin-up period) was 2.9% and 1.9% in the ten- and twenty-year REF inversions, respectively. Therefore, we note that the fully converged solution of

our inverse set-up will likely have higher interannual variability than the twenty-year inversion: whether this higher variability would be real or a result of overfitting of the available data (e.g. due to an underestimated model error) remains a topic of discussion. In both scenarios, however, the variability is typically smaller than 3%.

## 4 Further discussion

An important question concerns the robustness of the derived $[OH]_{GM}$ variations, as our inversion procedure did not provide formal posterior uncertainties. The ten-year inversions demonstrate that the derived solution over the 1998-2008 period is largely insensitive to the assumed prior OH and emission distributions (see Supplement S5), which we consider major drivers of uncertainty in the solution. The consistency between the three twenty-year inversions is weaker (e.g. Fig. 1), which we largely attribute to the convergence problems discussed in Section 3.5. However, especially after the prior MCF emissions stabilize in 2013, MCF emissions become a more dominant term in the MCF budget. An increased uncertainty in MCF emissions will likely affect the robustness of the solution, even if the strongly different spatial signatures of the OH distribution versus MCF emissions ensures some degree of spatial resolving power in the inverse framework. Therefore, we consider our solution most robust before 2008, and less robust after 2013, where it becomes more dependent on the limited resolving power of the inverse system.

Key to our inversions and the derived OH variations is the use of MCF observations from the NOAA-GML surface network. MCF has additionally been monitored by the Advanced Global Atmospheric Gases Experiment (AGAGE) surface network and its predecessors since 1978 (Prinn et al., 1983, 2018). Both networks maintain their own calibration scales for MCF, and previous comparisons have revealed site-dependent scale drifts and a possible phase difference between the two networks that exceed measurement repeatability (Holmes et al., 2013; Rigby et al., 2013). Due to these differences, and because of different site locations and measurement approaches, an inversion incorporating AGAGE instead of NOAA observations will result in different OH variations. It is difficult to assess the impact without performing additional inversions, but in a three-box inverse system differences between interannual OH variations derived from each network were shown to be insignificant in the presence of other uncertainties in the MCF budget (Rigby et al., 2017).

The only OH inversion studies that cover a similar period as this work were done in box models. As expected, the box model inversion that gave interannual $[OH]_{GM}$ variations most similar to those presented here is from our previous work (Naus et al., 2019), where we partly accounted for transport-related biases using TM5 ($R^2 = 0.49$; significant at p=0.01). This can be compared to the two-box inversion where we did not account for these biases ($R^2 = 0.21$; not significant at p=0.01). We find similarly low correlations with the OH variations derived in Turner et al. (2017) and in Rigby et al. (2017). In Section 3.3, we already noted that the two-box inversion where we accounted for transport biases also correlated better with the ENSO cycle, while the OH variations derived in this work correlate best with ENSO. This result, combined with the explicit simulation of transport and the more extensive use of available data (e.g. site-to-site gradients) give us confidence that the $[OH]_{GM}$ variations derived in this work are physically more realistic than those derived in two-box models. As such, we recommend the use of the $[OH]_{GM}$ variations derived in our twenty-year REF inversion as an alternative to interannually repeating OH fields, or to OH

fields derived in full-chemistry simulations, in for example CH$_4$ inversions. We do note that the [OH]$_{GM}$ variations derived in this work are statistically consistent with these previous studies, as uncertainties remain large relative to the amplitude of interannual variations in [OH]$_{GM}$.

It remains difficult to derive a multi-annual trend in [OH]$_{GM}$ from an inversion of MCF. To facilitate convergence of our inversion, we already pre-fitted MCF emissions to the global MCF trend. In principle spatially distinct patterns of emissions and OH still allow for an [OH]$_{GM}$ trend in the posterior solution, but the signal of such an [OH]$_{GM}$ trend in MCF observations would be small. Moreover, we have found evidence for a possible role of MCF release from oceans at high latitudes, as well as a growing relative contribution of MCF emissions in the MCF budget. While we can leverage [OH]$_{GM}$ variations as the most

likely source of interannual variations in the MCF growth rate, variations in these other budget terms become more important on decadal timescales. Therefore, on the basis of the inversions presented in this work, we conclude that a trend in [OH]$_{GM}$ cannot be excluded, though we have found no evidence for such a trend in the MCF budget.

Given the prominence of convergence problems in our results and our findings on the MCF ocean flux, some important recommendations for a future 3D model inversion of MCF can be inferred from this work. Most importantly, the prior MCF ocean

flux should include a high-latitude ocean source and this source would ideally be co-optimized with OH and MCF emissions, since our findings indicate that the intrahemispheric gradient of MCF informs on the ocean flux. Updated prior constraints on this ocean source in the form of lab measurements of MCF hydrolysis rates in cold water and field measurements would be highly valuable. In addition to an updated ocean flux, we think a reduction in the complexity of the inverse problem can greatly improve convergence. Firstly, the inversion can assimilate monthly-mean mole fractions of MCF, instead of individual observa-

tions, which we find difficult to reproduce in a global simulation. Secondly, OH could be optimized in fewer latitudinal bands, especially if the MCF ocean flux is co-optimized. Similarly, MCF emissions could either be optimized with a large-region approach, or with stiffer spatio-temporal correlations, for example by assuming slower variations in emissions ($> 1$ year). In addition to improved convergence, a reduced number of degrees of freedom in the observations and in the state vector makes other inverse approaches more feasible. For example, the problem can be solved analytically if the impact of perturbations in

each state element on observations is mapped, as in Bousquet et al. (2005). An improved 3D model inversion of MCF could build on our findings, but it will remain difficult to constrain global-scale OH variations of less than a few percent, even if such small variations can play a large role in the global CH$_4$ budget (e.g. Rigby et al., 2017).

While MCF has been the most widely used tracer for observation-derived constraints on [OH]$_{GM}$ over the past decades, promising alternatives have been explored. $^{14}$CO is produced from cosmic radiation in the stratosphere and subsequently

transported to tropospheric surface sites (Quay et al., 2000). Observed variations of $^{14}$CO could provide additional constraints on [OH]$_{GM}$ and the OH distribution, with a higher sensitivity to the tropics (Krol et al., 2008). Currently, long-term monitoring of $^{14}$CO is limited to one site (Baring Head, New Zealand), but efforts are being made to expand the network (Petrenko et al., 2019). Recently, it was proposed that high-resolution satellite data of CH$_4$ can be used to constrain both CH$_4$ sources and sinks, which would be possible with a combination of satellite products with different vertical sensitivities (Zhang et al., 2018).

Future research will reveal how feasible this approach is, but it will not provide information retroactively, for past decades: here, surface data still provides the strongest constraints. All potential tracers for OH are in principle complementary and the

implementation of multiple tracers in a comprehensive inversion framework could offer a more robust method for deriving OH variations (Liang et al., 2017). Already, we have shown that tracers such as $SF_6$ and HFC-152a can help in understanding the role of OH in the MCF budget. However, also in light of the results presented in this study, we deem it likely that each tracer will bring its own complications, which should be considered with care. This makes a multi-tracer approach a promising, but complicated and involving exercise.

Observational constraints have also been implemented in full-chemistry models. Assimilation of satellite data of those gases that dominate the OH budget, such as CO and $O_3$, is likely to bring full-chemistry simulations closer to reality and makes them less reliant on uncertain emission inventories (e.g., Miyazaki et al., 2012; Flemming et al., 2017). For example, assimilation of CO satellite data in a full-chemistry simulation resulted in an enhanced positive trend in $[OH]_{GM}$, compared to an unconstrained simulation, which was driven predominantly by a decreasing atmospheric CO burden (Gaubert et al., 2017). An approach complementary to budget-based inverse modelling and full-chemistry simulations has been suggested in formaldehyde (HCHO) (Wolfe et al., 2019). HCHO is well-monitored, both in-situ and by satellites, and the production of HCHO is tightly linked to OH abundance, especially in the remote troposphere. This approach provides high-resolution information on the OH distribution. All these methods benefit from the availability of each of the other methods as independent validation. Therefore, we consider that constraining the atmospheric oxidative capacity, a traditionally difficult problem at the heart of atmospheric chemistry, potentially has a solution in the convergence of a growing number of independent lines of evidence that need combined exploration.

## 5 Conclusions

In this study, we present a set of global 3D model inversions of MCF that cover the 1998-2018 period, performed within the TM5-4DVAR inverse framework. In the inversions, MCF emissions and OH concentrations were optimized simultaneously, with the objective to derive interannual variations in $[OH]_{GM}$ and in the OH distribution. The inverse system was typically able to reproduce observed MCF mole fractions within 1%. The main conclusions of this work can be summarized as follows:

1. Small interannual variability (2-3%) of $[OH]_{GM}$ without a significant trend allows for a relatively good match with MCF observations over the $1998-2018$ period. However, we also find a few larger multi-annual variations in $[OH]_{GM}$, such as an 8% increase from 2010 to 2012.

2. We have identified an underestimate of intrahemispheric gradients of MCF in the TM5 simulations that is not easily reduced using only adjustments in OH concentrations and anthropogenic MCF emissions. Instead, we propose that an explanation involving oceanic release of MCF at high latitudes can help to at least partly correct for these biases.

3. Most derived metrics, such as interannual $[OH]_{GM}$ variability and the absence of a trend in $[OH]_{GM}$, are relatively robust with respect to the prior [OH] and emission distribution. However, we put the most confidence in interannual variations of $[OH]_{GM}$, as trends and spatial adjustments of [OH] will be most strongly affected by remaining uncertainties in the system, such as the exact magnitudes of the oceanic flux and of anthropogenic emissions. Especially after 2013 the

role of emission uncertainties in our inverse system increases, as emissions seem to stabilize while MCF mole fractions continue to decline.

4. We find a significant and robust negative correlation between the derived variations in $[OH]_{GM}$ and the ENSO cycle. Such a correlation was not found in box model studies that did not fully account for transport variations.

5. Our optimized $[OH]_{GM}$ variations can add value to global $CH_4$ inversions, since the timeseries of optimized $[OH]_{GM}$ better reproduces MCF observations than annually repeating $[OH]_{GM}$. However, we emphasize that the solution remains very uncertain and the amplitude of derived $[OH]_{GM}$ variations depends on the degree of convergence.

*Data availability.* Optimized OH concentration and MCF emission fields derived in this study are available in the Supplemental Data. Additional results are available on request. Flask observations of MCF, $SF_6$ and HFC-152a from the NOAA surface network, as well as flask observations from the HIPPO campaign that were used in this study, are available on ftp://ftp.cmdl.noaa.gov/hats/. Flask observations from the ATom campaign can be retrieved from https://daac.ornl.gov/.

Literature estimates of the OH distribution, as shown in Figure S8, are available as follows: all estimates from the Atmospheric Chemistry and Climate Model Intercomparison Project (Lamarque et al., 2013) are available as Shindell et al.; OH fields from the global reananalysis of the Copernicus Atmospheric Monitoring Service (Inness et al., 2019) are available as CAMS reanalysis of chemical species - Hydroxyl Radical; OH fields from Lelieveld et al. (2016) are available as Gromov et al. (2020); OH fields from Gaubert et al. (2017) are available as Gaubert and Worden (2017).

*Author contributions.* MK, SN and SM designed the research. SN wrote the manuscript with major input from MK, and further contributions from all co-authors. SN performed the TM5 inversions and analysed the results. MK supervised the research. All authors discussed the results and contributed to the final manuscript.

*Competing interests.* The authors declare that they have no conflict of interest.

*Acknowledgements.* This work was carried out on the Dutch National e-Infrastructure with the support of SURF Cooperative. This work was funded through the Netherlands Organisation for Scientific Research (NWO), project number 824.15.002. Maarten Krol received funding from the European Research Council (ERC) under the European Union's Horizon 2020 research and innovation programme under grant agreement No 742798. Prabir K. Patra is partly supported by Environment Research and Technology Development Fund (#2-1802) of the Ministry of the Environment, Japan. We thank the NOAA and cooperative station personnel for diligence and care in the collection of flasks. We thank C. Siso and B. Miller for assistance with analysis of ground-station flasks and flasks collected during aircraft campaigns, F. Moore and E. Atlas for assistance with flask sampling during HIPPO and ATom, and the broader NSF and NOAA teams for enabling and facilitating those projects.

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
