# Peer review of "A 3D-model inversion of methyl chloroform to constrain the atmospheric oxidative capacity"

_Atmospheric Chemistry and Physics, 2020_

## Referee Comment (RC1) · Anonymous Referee #1 · 5 Sep 2020

General comments: This paper conducted a 3D model inversion of global MCF emissions and OH concentrations for 1998-2018. The inversion results show that the interannual variations of OH are small and there is no significant long-term trend in OH. If the main conclusion is correct, it can be an important contribution to our understanding of the global CH4 budget. Overall, the paper is addressing important questions and within the scope of ACP. However, some questions need to be clarified. I would like to recommend the publication of the manuscript if the following questions are addressed.

Major comments: 1) The inversion set-up. Unlike previous studies (e.g. Bousquet et al. (2005)) that together optimized MCF destructions by OH and the ocean sink, this study only optimized OH and applied the first-order ocean flux. Thus the inversion results may be largely impacted by the uncertainties in the prescribed ocean flux. Before inver-

sion, the prior emissions were pre-optimized to fit the global mean MCF mole fractions. It is not clear why the emission needs to be pre-optimized. Will the pre-optimization reduce the OH variation estimated by the inversions since the MCF emissions already fit the observations? The study assumed a 50% error for MCF emissions. The MCF emissions become small after 1997. In Turner et al. (2017), the error for the MCF emission in the northern hemisphere is set to no less than 1.5 Gg/y. Will the assumption that prior MCF emission error proportional to emissions lead to underestimation of the prior error?

2) The optimized OH is not well presented in the results. Only the interannual variations weighted by temperature were shown in Figure 1 and Figure 2. For the posterior OH, what are the global tropospheric mean OH concentrations and the corresponding CH4 lifetime? Are the latitudinal distributions consistent with previous studies? What is the N/S ratio? In the inversion TM5OH, the prior OH field shows higher concentrations over the northern hemisphere. Is the inversion using two different prior OH distribution estimated similar posterior latitudinal OH distributions? I think these values are also worth discussing.

3) The inversion results show small OH interannual variability and no significant OH trend. This is different from previous top-down studies using the two-box model inversion (Rigby et al., 2017; Turner et al., 2017). The author shows that the interannual variation can be supported by the negative correlation with the ENSO cycle. The large negative OH anomaly during 1998 (El Niño year) has been proven by several previous studies (Bousquet et al., 2006; Butler et al., 2005; Nguyen et al., 2020; Zhao et al., 2020). However, here the inversion shows a large positive anomaly in 1998. Besides, how to explain is a large positive anomaly in 2012?

Other comments: L75: What is meant by "the most promising period in its measured history"?

L110: Are the interannual variations of stratospheric photolysis considered over the

inversion time period?

L118: The 10% error in latitudinal OH may be underestimated. Usually, the error for global annual mean OH is given by ∼10%. But for the monthly mean OH averaged for 5° latitude, the error can be much larger (e.g. Naik et al., 2013; Zhao et al., 2019).

L105-115: How you get the initial conditions for MCF? Are the initial conditions also optimized?

L128-L133: Are the MCF emissions also pre-optimized in POP and TM5OH?

L175: The results during the spin-up and spin-down period are not significant, I suggest the author remove the results of the corresponding period.

L196: The top-down estimated emissions and OH variations also depend on the variations of observed atmospheric MCF concentrations and the reaction rates (temperature) of MCF with OH. It is not clear for me why the small correlation coefficient between OH and MCF variations can reflect the OH and MCF are independently derived.

Figure 4: The MCF emissions are pre-optimized to reproduce the global mean MCF mole fractions, why there are still very large mismatches between the model simulated and observed MCF mixing ratios (dash line)?

In Figure 3, Figure 4, and Figure 5, the line color corresponding to each inversion experiment is different. I suggest using the same color for each experiment in different figures.

L254-L255: Is the bias in OH vertical distributions also contribute to the model-measurement mismatch?

L277: Why MEI should lead by one year? Is this mean that the OH should show a negative anomaly one year after the El Niño? This is not consistent with the explanation in L283-L286 and previous studies (Nguyen et al., 2020; Zhao et al., 2020), which show negative OH anomaly during El Niño years.

L349: What are the criteria for stopping the iteration? Are the 3 inversion experiments reach a similar value of cost function J (and the gradient of the cost function) in the last iteration?

L352: Why further convergence will result in less realistic OH variations?

L372: "Firstly...we generally identified similar tendencies in each." Figure 1 has shown the variations of OH estimated by three inversions are quite different.

L374-379: Here the manuscript tried to prove the robustness of the OH interannual by an additional inversion and a forward simulation. But the details of the two experiments and the results are not given. I suggest include some details in the supplements. E.g. how the one global scaling factors compare with the REF, POP, and TM5OH? Is the forward simulation use the prior or optimized MCF emissions? I cannot understand the logic here, can you clarify why the two experiments can indicate the robustness of the derived OH variations?

Reference Bousquet, P., Hauglustaine, D. A., Peylin, P., Carouge, C., and Ciais, P.: Two decades of OH variability as inferred by an inversion of atmospheric transport and chemistry of methyl chloroform, Atmos. Chem. Phys., 5, 2635-2656, 10.5194/acp-5-2635-2005, 2005. Bousquet, P., Ciais, P., Miller, J. B., Dlugokencky, E. J., Hauglustaine, D. A., Prigent, C., Van der Werf, G. R., Peylin, P., Brunke, E. G., Carouge, C., Langenfelds, R. L., Lathiere, J., Papa, F., Ramonet, M., Schmidt, M., Steele, L. P., Tyler, S. C., and White, J.: Contribution of anthropogenic and natural sources to atmospheric methane variability, Nature, 443, 439-443, 10.1038/nature05132, 2006. Butler, T. M., Rayner, P. J., Simmonds, I., and Lawrence, M. G.: Simultaneous mass balance inverse modeling of methane and carbon monoxide, Journal of Geophysical Research: Atmospheres, 110, 10.1029/2005jd006071, 2005. Rigby, M., Montzka, S. A., Prinn, R. G., White, J. W. C., Young, D., O'Doherty, S., Lunt, M. F., Ganesan, A. L., Manning, A. J., Simmonds, P. G., Salameh, P. K., Harth, C. M., Muhle, J., Weiss, R. F., Fraser, P. J., Steele, L. P., Krummel, P. B., McCulloch, A., and Park, S.: Role of atmospheric oxidation in recent methane growth, Proc Natl Acad Sci U S A, 114, 5373-5377, 10.1073/pnas.1616426114, 2017. Turner, A. J., Frankenberg, C., Wennberg, P. O., and Jacob, D. J.: Ambiguity in the causes for decadal trends in atmospheric methane and hydroxyl, Proc Natl Acad Sci U S A, 114, 5367-5372, 10.1073/pnas.1616020114, 2017. Naik, V., Voulgarakis, A., Fiore, A. M., Horowitz, L. W., Lamarque, J. F., Lin, M., Prather, M. J., Young, P. J., Bergmann, D., Cameron-Smith, P. J., Cionni, I., Collins, W. J., Dalsøren, S. B., Doherty, R., Eyring, V., Faluvegi, G., Folberth, G. A., Josse, B., Lee, Y. H., MacKenzie, I. A., Nagashima, T., van Noije, T. P. C., Plummer, D. A., Righi, M., Rumbold, S. T., Skeie, R., Shindell, D. T., Stevenson, D. S., Strode, S., Sudo, K., Szopa, S., and Zeng, G.: Preindustrial to present-day changes in tropospheric hydroxyl radical and methane lifetime from the Atmospheric Chemistry and Climate Model Intercomparison Project (ACCMIP), Atmospheric Chemistry and Physics, 13, 5277-5298, 10.5194/acp-13-5277-2013, 2013. Nguyen, N. H., Turner, A. J., Yin, Y., Prather, M. J., and Frankenberg, C.: Effects of Chemical Feedbacks on Decadal Methane Emissions Estimates, Geophysical Research Letters, 47, e2019GL085706, 10.1029/2019gl085706, 2020. Zhao, Y., Saunois, M., Bousquet, P., Lin, X., Berchet, A., Hegglin, M. I., Canadell, J. G., Jackson, R. B., Hauglustaine, D. A., Szopa, S., Stavert, A. R., Abraham, N. L., Archibald, A. T., Bekki, S., Deushi, M., Jöckel, P., Josse, B., Kinnison, D., Kirner, O., Marécal, V., O'Connor, F. M., Plummer, D. A., Revell, L. E., Rozanov, E., Stenke, A., Strode, S., Tilmes, S., Dlugokencky, E. J., and Zheng, B.: Inter-model comparison of global hydroxyl radical (OH) distributions and their impact on atmospheric methane over the 2000–2016 period, Atmos. Chem. Phys., 19, 13701-13723, 10.5194/acp-19-13701-2019, 2019. Zhao, Y., Saunois, M., Bousquet, P., Lin, X., Berchet, A., Hegglin, M. I., Canadell, J. G., Jackson, R. B., Deushi, M., Jöckel, P., Kinnison, D., Kirner, O., Strode, S., Tilmes, S., Dlugokencky, E. J., and Zheng, B.: On the role of trend and variability of hydroxyl radical (OH) in the global methane budget, Atmos. Chem. Phys. Discuss., 2020, 1-28, 10.5194/acp-2020-308, 2020.

[Figure]

2020.

---

## Referee Comment (RC2) · Anonymous Referee #2 · 10 Sep 2020

The paper presents an estimate of methyl chloroform (MCF) tropospheric loss and emissions using a global 3D model and a 4DVAR data assimilation system. The subject is important, because MCF has been used extensively, mostly in 2D box models, to estimate the magnitude and variability of global mean concentrations of the hydroxyl radical (OH), which is responsible for removing gases such as methane from the atmosphere.

The work is timely and it is important, given the current uncertainty surrounding the global methane budget and the fate of other atmospheric constituents. I was very much hoping that this paper would provide a valuable new constraint on this complex problem, because the model and inverse approach is well established. There is a lot to like in this paper, and in many respects, the authors have done a very thorough

investigation into the solution they have found.

However, I am concerned about the stated lack of convergence in the main results. I believe that this substantially reduces the confidence we can have in the results and conclusions and implies that the main results are not reproducible. This lack of convergence is investigated in Section 3.5, and it is stated that of the three main inversions exhibited a different level of convergence. The presented results are therefore somewhere between the prior and posterior solution, but we don't know where. For some of the inversions, the "main" results are substantially different from the converged posterior, as shown for a subset of years in Figure S6. Put another way, the solutions presented are not consistent with the data, the cost function and the prior assumptions. Of course, if my understanding is correct (and please clarify it not), this would imply that the results of this paper would not be fully reproducible, even if I used the same data, model, cost function and priors.

Of further concern, the reason given for presenting a partially converged solution is that the fully converged solution does not look physically realistic. I think there are at least two problems with this reasoning. Firstly, it is not clearly articulated why these solutions are necessarily unphysical (large changes to the priors are identified, but it is not stated why these cannot occur). Secondly, it is surely not a valid approach to prematurely stop your inversion midway down its descent because you don't like what you see at the bottom. Wouldn't the preferable approach be to interrogate the physical model, the observations or the uncertainty assumptions, to understand why the converged solution is the way it is and try to formulate an alternative which provides a more physically reasonable result?

The authors do present a set of solutions which have converged, but only for a subset of years (10 years, as shown in Figure S6). If no further runs of the model are possible for the full 20 year period, I suggest the paper be refocused only on those years where a converged solution is found. To me, it would seem preferable to present a converged solution over a shorter time period, than a longer solution that could is not consistent

with the data, model and uncertainty assumptions.

As I said, there is a lot to like in this paper, and I hope the authors can fix these convergence issues and resubmit.

I have some other general points, which I hope are helpful:

1. The importance of using ocean fluxes that account for absorption and reemission, compared to a 1st order loss has been well known for almost 20 years, at least for the overall MCF trend, and particularly during the period where emissions were changing rapidly. This article presents a nice demonstration of the influence of different ocean flux parameterizations on the meridional gradient. However, given that it is well established, I'm puzzled as to why the more realistic ocean fluxes weren't used in the main inversions? It would surely be preferable to rerun the inversions using a more accurate ocean flux estimate, precisely because, as the authors show, they can influence the solution in important ways. Further, as an aside, the type of ocean flux parameterization used in the inversions (i.e. 1st order loss) does not seem to be explained in the main text but needs to be specified.

2. A main conclusion of the paper is that the variation in oxidation magnitude is small (< 3% per year). This does indeed seem to be the case from the point of view of the standard deviation in the solution. However, some year-to-year changes in fact seem to be very large. For example, sometime around 2010 − 2012, the REF inversion shows a change from ∼-5% to ∼+5% compared to the prior (Figure 1). Wouldn't a change in tropospheric oxidation of 10% over 2 years actually be considered quite substantial, and have major impacts on, for example, the global methane budget?

An additional point: converged solutions in Figure S6 seem to show, in general, more variation than the unconverged "main" results. So, again, it would be important to investigate more fully how sensitive this main conclusion is to the lack of convergence in the main results.

3. If emissions are being derived in the main inversions, why was it necessary to "pre-optimize" the emissions, assuming constant loss? What happens if you don't do this? If this changes the result substantially, I'd be very concerned, as you're essentially using the observations twice, and, in the first step, you're fixing one of the parameters that you are trying to infer in the second pass. If it doesn't change the results substantially, then wouldn't this step be unnecessary?

Minor comments

L7: ". . . better reproduce. . ." (than what?)

L18: "the signals are small". This is too vague. Need to specify what quantities are being referred to.

L19: ". . . better match the global MCF observations. . ." (than what?)

L33 − 35: citation needed.

L48: What does the phrase: "but artifact-free sampling has become more difficult" mean?

L91 and throughout: phrases like "such as OH" need to specify that it is OH concentration that is being referred to. Perhaps define "[OH]".

L155: Can you be more specific than saying that model error was "proportional to the 3D spatial gradients". I.e. define what you mean by spatial gradients and let us know if there was a constant of proportionality.

L180: ". . . correlation with the REF inversion" (need to state what the correlation is with respect to)

L195: I'm not sure what you mean by "would be hard to exclude from a bottom-up perspective". Can you be more explicit?

Figure 4: There seems to be a consistent increase in the mismatch in the IH gradient

around 2013. Any idea what causes this? Seems like a potentially interesting feature. Could this point to a sudden increase in emissions?

Figure 5 and throughout: For the reader not familiar with these site codes, it would be useful to clarify where these stations are (i.e. their latitude is particularly important for the discussion of Figure 5).

L374-375: It would seem important to include the results of this test (scaling global OH) in the supplement.

L379: "These results indicate substantial robustness of the derived OH variations". I'm wondering if this can really be stated so strongly, given that the three main inversion show different OH variations?

---

## Author Comment (AC1) · 23 Oct 2020

We thank the reviewer for the insightful comments. Most comments concern the configuration of the inversions. Often, we agree with these comments. Based on the lessons we have learned during our work on the inversions presented in this manuscript, we would design a new set of inversions differently. In this response we would like to make two points:

1. We did run test inversions and we did consider all relevant previous work when we designed our inversions, and so we designed these inversions as best as we could, with the information available.

2. We have learned valuable lessons from our work that we consider worth sharing with the scientific community.

Below, we discuss the reviewer's comments in more detail.

1) The inversion set-up. Unlike previous studies (e.g. Bousquet et al. (2005)) that together optimized MCF destructions by OH and the ocean sink, this study only optimized OH and applied the first-order ocean flux. Thus the inversion results may be largely impacted by the uncertainties in the prescribed ocean flux.

We agree with the reviewer that the ocean sink can play an important role in the MCF budget, as becomes apparent also in our manuscript. Our treatment of the ocean sink is based on pragmatic considerations. During test inversions we already diagnosed possible convergence problems. The inverse system is very delicate and to derive a physical realistic solution we had to carefully balance all the terms in the cost function. Therefore, in the design of this study, we considered the expected difficulty of co-optimizing an ocean sink to be more important than the possible advantage of including it.

In our manuscript, we do consider the impact of this choice. Partly, because we show that using an ocean sink different from the one in our standard inversion can have a significant impact on the latitudinal distribution of MCF, for example in Supplement S3. We also discuss the potential impact of an uncertain ocean sink on our results in Lines 420-427 of the discussion. To summarize the results, the impact on the derived interannual variability of OH will likely be small, but the trends and latitudinal distribution of OH could be affected more significantly. Another reason to not optimize the ocean sink is that we do not think that the NOAA surface network can separate destruction by OH from a Wennberg-like ocean flux, since both have a similar latitudinal distribution.

The reviewer refers to Bousquet et al. (2005) for an example where co-optimization of an ocean flux does work out. The inverse system employed in that study differs significantly from the one we use, and consequently it has different strengths and limitations. Whereas we calculate a comprehensive cost function gradient in each iteration with the adjoint model, they approximated the sensitivity of monthly mean MCF observations to a limited number of state variables in a set of forward simulations. This makes their inverse estimation computationally more efficient, because once the source functions are calculated, the 3D transport model doesn't need to be run anymore. On the other hand, we were able to optimize grid level emissions of MCF, as well as OH in 45 latitudinal bands, with individual surface observations. The source-function approach necessitates a large-region approach with monthly mean observations to remain computationally viable. In fact, one recommendation in Bousquet et al. (2005) was to move away from this large-region approach, which we have done.

In hindsight, we are not sure which approach works best. The increased complexity of the inverse problem that we tried to solve is likely an important reason that we ran into convergence problems. The added value of individual observations over monthly mean observations is questionable, and the sourcefunction approach is an elegant solution to the computational expense of a multi-decadal inversion. However, these are lessons we have learned that we want to communicate to the science community.

We extensively discuss the convergence problems in our manuscript and we have tried to assess the possible impact on our results, for example by reducing the problem to a ten-year inversion (Supplement S4). Based on all of our findings, we remain confident that the timeseries of OH variations we derive in the REF inversion is credible and consistent with the MCF observational record.

Before inversion, the prior emissions were pre-optimized to fit the global mean MCF mole fractions. It is not clear why the emission needs to be pre-optimized. Will the pre-optimization reduce the OH variation estimated by the inversions since the MCF emissions already fit the observations?

The study assumed a 50% error for MCF emissions. The MCF emissions become small after 1997. In Turner et al. (2017), the error for the MCF emission in the northern hemisphere is set to no less than 1.5 Gg/y. Will the assumption that prior MCF emission error proportional to emissions lead to underestimation of the prior error?

Firstly, we would like to correct an error in the manuscript. The assumed gridbox error is 50% over 1998-2005, but increases with 15% per year afterwards, up to a maximum of 200% after 2015. This is partly to reflect growing uncertainties in emission inventories and partly because emissions become exceedingly small in later years, as is indicated by the reviewer. Our minimum prior MCF emissions are around 2 Gg/year (e.g. Fig. 3), so the minimum emission uncertainty is 4 Gg/year, on the high end of what has been assumed in previous work. We have now outlined our prior settings correctly in the manuscript (Lines 120-123).

Initially, we ran inversions without pre-fitted emissions and assumed minimum emissions of 5 Gg/year. The result was a large overestimation of MCF mole fractions in later years. Therefore, many expensive iterations were required to even get close to observed mole fractions. We use a 3D transport model because we are interested in the subtle spatial gradients between surface sites, the temporal variations therein, and if and how these gradients inform on the separation between MCF emissions and OH. Pre-fitting the global MCF emissions to global mean MCF mole fractions turned out an efficient ways to reduce computational costs.

The impact of uncertain MCF emissions on derived OH variations is somewhat similar to that of the ocean sink discussed above: more likely to impact an OH trend than interannual variability of OH. However, the difference with the ocean sink is that we have found evidence that the inversion can spatially separate OH from MCF emissions to some degree. For example, in the REF inversion, we derive OH and MCF emission variations that are uncorrelated, signaling different degrees of freedom in the optimization.

The optimized OH is not well presented in the results. Only the interannual variations weighted by temperature were shown in Figure 1 and Figure 2. For the posterior OH, what are the global tropospheric mean OH concentrations and the corresponding CH4 lifetime? Are the latitudinal distributions consistent with previous studies? What is the N/S ratio? In the inversion TM5OH, the prior OH field shows higher concentrations over the northern hemisphere. Is the inversion using two different prior OH distribution estimated similar posterior latitudinal OH distributions? I think these values are also worth discussing.

We agree that there are many aspects of the derived OH distributions that are interesting. This is why we decided to provide the derived distributions in various formats in the Supplemental Data and urge any follow-up study to carefully consider which of these metrics best fits their purpose. We now provide the tropospheric lifetimes of MCF and $CH_4$ with respect to oxidation by OH in the manuscript and a

timeseries of these quantities in Supplement 1. We find that the derived lifetimes agree well with literature estimates.

The N/S ratio and global mean OH concentration are a reference quantities used often in previous literature. However, in the context of our study, we do not consider them informative and possibly misleading. We have found some adjustment to the N/S ratio and large adjustments in the global mean OH concentration, but this is a side-effect of the very large displacement of extra-tropical OH towards the tropics. This displacement of OH we deem unrealistic in amplitude and we at least partly attribute it to an erroneous ocean sink. Consequently, from the results presented in this study we are not confident that we can draw any conclusions regarding a N/S ratio or the global mean OH concentration and we are therefore hesitant to provide updated estimates.

We have changed Supplemental Figure S8, so that it now includes a comparison of our prior and posterior latitudinal distributions of OH with literature estimates, mostly from full-chemistry simulations. Here the difference between the TM5-OH and the Spivakovsky OH distributions can be clearly observed. Additionally, it is clear that the adjustments made to the prior distribution of OH are similar for the three different inverse set-ups in the ten-year inversions, although differences that were present in the prior OH distributions (e.g. the double peak in TM5-OH) are not adjusted.

The inversion results show small OH interannual variability and no significant OH trend. This is different from previous top-down studies using the two-box model inversion (Rigby et al., 2017; Turner et al., 2017). The author shows that the interannual variation can be supported by the negative correlation with the ENSO cycle. The large negative OH anomaly during 1998 (El Niño year) has been proven by several previous studies (Bousquet et al., 2006; Butler et al., 2005; Nguyen et al., 2020; Zhao et al., 2020). However, here the inversion shows a large positive anomaly in 1998. Besides, how to explain is a large positive anomaly in 2012?

Firstly, while we do not find evidence for a trend in OH, we also do not find evidence for the absence of a trend. This strongly relates to the uncertain ocean sink and MCF emissions, which we discuss above and in Lines 420-427 of the manuscript. Similarly, neither of the two-box model studies mentioned was able to exclude constant OH throughout their inversion period, so we consider our results to be consistent with these studies.

Top-down modeling studies of MCF that constrain OH have traditionally not provided strong evidence for physical mechanisms that could drive the derived OH variations. Here, we show that one strong driver of atmospheric variability (ENSO) correlates well with our derived OH variations. We consider this an improvement on previous work.

The positive OH anomaly in 1998 is likely due to spin-up effects. We mention this in the manuscript the first time we introduce global OH variations:

*We have shown the entire twenty-year inversion period, which will include a spin-up and spin-downperiod of 1-2 years (indicated by the gray bars). For example, even though our initial MCF mole fraction fields are realistic, the strong positive oxidation anomaly in 1998 might be linked to errors in the initial field.* Lines 182-184

We now also exclude 1998 and 2017 from our comparison with ENSO and indicate spin-up and -down periods in Figures 1, 2 and 8 as gray shaded areas.

We do not know what could drive the 2012 positive anomaly, but we also do not consider OH chemistry to be well-enough understood to exclude an anomaly of a few percent.

Other comments: L75: What is meant by "the most promising period in its measured history"?

Our message was that we investigate the period where MCF emissions are low, but MCF mole fractions (and thus loss to OH) are still relatively high. We have changed the phrasing:

*The objective of this study is to investigate information on large-scale [OH] variations contained in measurements of the most promising tracer identified to date, MCF, during the period that follows its drop in emissions, using the most comprehensive tools available to us, in the form of a state-of-the-art inverse system built around a 3D transport model.* L80

L110: Are the interannual variations of stratospheric photolysis considered over the inversion time period?

Interannual variations in the stratospheric photolysis rates are not considered. However, due to a changing stratospheric burden and distribution of MCF, and due to interannual variations in stratosphere-troposphere exchange, the absolute photolysis sink will vary. While in reality stratospheric photolysis rates can also vary interannually, we consider this effect second-order to other budget terms, similar to previous MCF inversions.

L118: The 10% error in latitudinal OH may be underestimated. Usually, the error for global annual mean OH is given by 10%. But for the monthly mean OH averaged for latitude, the error can be much larger (e.g. Naik et al., 2013; Zhao et al., 2019).

It is difficult to define a prior error on OH: it does not really represent the error on monthly latitudinal averages, because we have included strong temporal and spatial correlations. It is possible to argue that we should have chosen a larger error, but in the current set-up the inversion still finds adjustments to the prior OH distribution of up to 50%. This indicates that the prior and observational errors should be interpreted as balancing terms of the different components in the cost function, rather than as quantities that can be interpreted in an absolute sense.

Given the large amplitude of adjustments that we find, we consider that the cost function is well-balanced between observations and prior, or if it is unbalanced that we have attached too much weight to the observations (i.e. we are overfitting). Therefore, we do not think a larger error on OH would be beneficial to the inversion: if anything we would consider increasing the observational error.

L105-115: How you get the initial conditions for MCF? Are the initial conditions also optimized?

Initial fields were obtained from a 1988-1998 forward simulation that used initial fields, sources and sinks from the TransCom-CH4 protocol. As discussed in Patra et al. (2011), MCF fields in these simulations match observations quite well. The initial field was not optimized. Instead, the first 1.5 years of the inversion are considered as spin-up period during which optimized emissions and OH might correct for errors in the initial MCF distribution.

L128-L133: Are the MCF emissions also pre-optimized in POP and TM5OH?

Yes, and additionally we use the same global emission totals for all inversions. For example:

*In our second inversion, referred to as POP, we redistributed the same annual total MCF emissions as in the REF inversion proportional to population density..* L134-145

L175: The results during the spin-up and spin-down period are not significant, I suggest the author remove the results of the corresponding period.

We have chosen to still include the spin-up and spin-down period in the revised manuscript. Instead, we have demarcated these periods as shaded areas in Fig. 1, 2, 8. The reason is that we do not know exactly how to quantify the length of these periods and so we consider it impossible to exclude a sufficiently long spin-up/down period without risking that we exclude potentially interesting information. We leave it now to the discretion of the reader to interpret and to use our timeseries correctly. We hope that in the revised manuscript it is clear that for example the positive 1998 OH anomaly is more likely to be a spin-up artifact than a real OH variation.

L196: The top-down estimated emissions and OH variations also depend on the variations of observed atmospheric MCF concentrations and the reaction rates (temperature) of MCF with OH. It is not clear for me why the small correlation coefficient between OH and MCF variations can reflect the OH and MCF are independently derived.

We can consider the extreme of a one-box model inversion of MCF that optimizes OH and emissions. In this set-up, OH and MCF emissions have opposite but indistinguishable effects on the modeled quantity: global mean MCF. Therefore, if simulated MCF is too low this can be compensated partly by a decrease in OH and partly by an increase in emissions. Depending on the error settings, the result will be a strong negative correlation between the posterior adjustments to emissions and OH.

In our 3D inversion, observational constraints from spatial MCF gradients are included and so there is some skill to separate OH from emissions, since their respective spatial distributions are very different. However, the surface network is sparse and since MCF has a long lifetime relative to most atmospheric transport timescales the distinction between emissions and OH is not sharp. Therefore, it is difficult to predict a priori if and how well this separation will work.

This is why we report the correlation between derived adjustments to emissions and OH. If the correlation would be strongly negative, then this is evidence that emissions and OH are used to correct for the same model-measurement differences, as expected in the one-box model. That this is not the case provides some evidence that the OH and emission adjustments address different aspects of the MCF gradients.

Figure 4: The MCF emissions are pre-optimized to reproduce the global mean MCF mole fractions, why there are still very large mismatches between the model simulated and observed MCF mixing ratios (dash line)?

For pre-optimization of emissions we needed to convert observed, global mean mole fractions to an atmospheric burden. For this we needed to define a global total atmospheric mass, which is a parametrized quantity, since emissions initially mainly spread through the troposphere and only later to the stratosphere. Apparently, we choose a somewhat too low atmospheric mass, resulting in too-low mole fractions in the REF and POP inversions.

We could have adjusted the emissions upwards, but we actually considered the poor match a good test of the inverse system. It shows that we can fit global mean mole fractions of MCF, when the prior mismatch is within reasonable bounds (which was not the case in the test inversion where we floored the emissions at 5 Gg/year; see above). Therefore, we decided that these pre-optimized emissions are a good starting point for the optimizations.

In Figure 3, Figure 4, and Figure 5, the line color corresponding to each inversion experiment is different. I suggest using the same color for each experiment in different figures.

We agree and have adjusted the colors to be consistent.

L254-L255: Is the bias in OH vertical distributions also contribute to the model-measurement mismatch?

This is a good point, we make mention of it now in the manuscript:

*Estimates of the total atmospheric oxidizing capacity are more likely to be affected by a systematic underestimate of vertical MCF gradients. This underestimate could be driven by an underestimate of the vertical OH gradients or by too-fast vertical mixing in TM5.* L277-279

L277: Why MEI should lead by one year? Is this mean that the OH should show a negative anomaly one year after the El Niño? This is not consistent with the explanation in L283-L286 and previous studies (Nguyen et al., 2020; Zhao et al., 2020), which show negative OH anomaly during El Niño years.

We agree that this lag was an unnecessary addition. We now only consider a zero-lag correlation for the 1999-2016 period (i.e. excluding one year spin-up and spin-down) and have slightly rephrased this section.

**References**

1. Bousquet et al. (2005), Two decades of OH variability as inferred by an inversion of atmospheric transport and chemistry of methyl chloroform, *Atmos. Chem. Phys., 5, 2635–2656*
2. Patra et al. (2011), TransCom model simulations of $CH_4$ and related species: linking transport, surface flux and chemical loss with $CH_4$ variability in the troposphere and lower stratosphere, *Atmos. Chem. Phys., 11, 12813–12837*

---

## Author Comment (AC2) · 23 Oct 2020

We thank the reviewer for the kind words and for the good suggestions to present the results in the manuscript more convincingly.

The main concern is by the reviewer named "lack of convergence" in our solution that includes a twenty-year timeseries of global OH variations. We agree with the reviewer that the degree of convergence is an issue of concern, which is why we treat it separately in Section 3.5. Our study has been constrained mostly by pragmatic concerns. To give an indication of the computation time involved: to reach the degree of conversion presented in the manuscript, around thirty wall-clock days were needed. The set-up of the final inversion was inspired by test inversions and, likely, more improvements in this set-up are possible. We expect, however, that new challenges will emerge if we implement these improvements. We decided to stop this cycle and share our findings with the research community. With these inversions we believe that we have reached an acceptable endpoint, which we will argue further here.

In the statistical framework that we have defined, the solution has indeed not fully converged: a smaller cost function would be found, if we would continue to iterate. We have stopped the inversion for two main reasons:

1) **The standard REF solution presented in the manuscript reproduces NOAA surface observations to a satisfying degree.** Although we do not reproduce each individual observation within the error we have prescribed, we suspect that the pre-defined observational + modeling error might have been too small. We discuss this issue in Lines 246-249 of the manuscript. It is difficult to formulate a correct modeling error a priori, because MCF gradients are small and sub-grid variations can be important. That it is difficult to define such an error can be seen in Bousquet et al. (2005), where the modeling error is actually defined in hindsight, to reach a chi-squared of 1. We cannot use the same approach due to computational expenses in our set-up, but we extensively investigate what causes the residual errors and we consider the residuals small enough that we can call our REF solution a satisfactory solution. To put it differently: given the fact that we might have underestimated the model error on observations, iterating further towards a fully converged solution could be considered overfitting.

   In hindsight, our recommendation would be to not assimilate individual observations, but instead assimilate monthly means, because with relatively homogeneous OH and emission fields we cannot reproduce individual observations anyway (lines 235-236). However, since we already reproduce monthly means at most sites, we do not consider implementation of this recommendation a prerequisite for publication. We have included this recommendation in the new version of the manuscript (lines 250-251).

2) **Further iterations result in a lower cost function, but not necessarily in a better solution.** The residual cost function appears to be dominated by intrahemispheric biases and by short-term variations. As discussed above and in lines 243-244, there are simply not enough degrees of freedom in the inverse framework to improve the match with short-term variations.

   The intrahemispheric bias is a different concern. Already, adjustments in the latitudinal OH distribution in the 20-year REF inversion are up to 30%, which corresponds to 3-sigma. In the better-converged 10-year inversion, we find adjustments of 60%, or 6-sigma. As discussed in the manuscript, we consider it unlikely that these adjustments in OH provide the best solution to explain the intrahemispheric biases, but it is the only explanation our inversion can provide us with, in this set-up.

   The reviewer accurately notes that we did not provide convincing evidence that such adjustments are physically unrealistic. We have modified Figure S8, that now compares literature estimates of OH distributions to our prior and posterior OH distributions. It can be observed that, in the ten-year inversions, which came closest to reproducing the intrahemispheric gradients, the ratio between tropical and extra-tropical OH ends up much higher than in estimates from a range of chemistry models. Although we cannot exclude the possibility that tropical OH is too low in the prior distribution, we derive adjustments far outside the (arguable uncertain) prior error settings. One of our noteworthy hypotheses is therefore that a scenario that includes a high-latitude ocean source of MCF is more likely than one that does not.

In summary, we agree with the reviewer that stopping an inversion half-way is suboptimal and statistically inconsistent. However, we emphasize in the manuscript that the error we put on observations might be too small and that our inversion reproduces NOAA surface observations well at most sites. Additionally, where the agreement is poor, we do not think that more iterations will help.

Further general comments:

1. The importance of using ocean fluxes that account for absorption and reemission, compared to a 1st order loss has been well known for almost 20 years, at least for the overall MCF trend, and particularly during the period where emissions were changing rapidly. This article presents a nice demonstration of the influence of different ocean flux parameterizations on the meridional gradient. However, given that it is well established, I'm puzzled as to why the more realistic ocean fluxes weren't used in the main inversions?

We do not agree that the ocean flux described in Wennberg et al. (2004) would a priori have been a better choice than the simple first-order loss we have used. In Supplement S4, we outline the observational evidence for the low hydrolysis rates at cold temperatures. These low hydrolysis rates are key to the hypothesis of an ocean source, but the evidence is thin: it is based on extrapolation of hydrolysis rates measured above 25°C, and the only study performed at 10°C found higher-than-expected hydrolysis rates. We are not aware of any experimental follow-up studies. We have included a recommendation in the supplement (lines 122-124).

In Rigby et al. (2017) it is argued that inversions of MCF that do not adopt an ocean source derive spurious OH variations particularly around 1998. However, we would argue that during this period uncertainty in emission timing (particularly delayed emissions) could equally well explain the derived variations. Large changes in MCF emissions coincide with the hypothesized onset of an ocean source, which is why we have found it impossible to find evidence for a switch in sign of the ocean flux in the surface network observations during this time. On that note, in Naus et al. (2019), we have performed a two-box inversion of MCF, covering 1994-2014, that included a first-order ocean sink and we were able to reproduce hemispheric averages of MCF without large OH variations.

In the absence of strong evidence for (or against) an ocean source, we choose to adopt the simplest assumption: first-order ocean loss. Having performed our 3D model inversions, we think that we have found convincing evidence in the latitudinal gradients of the surface networks for an ocean source of MCF. Given that we do not consider an ocean source well-established, we present this as one of the main findings of our study.

2. A main conclusion of the paper is that the variation in oxidation magnitude is small (< 3% per year). This does indeed seem to be the case from the point of view of the standard deviation in the solution. However, some year-to-year changes in fact seem to be very large. For example, sometime around 2010 – 2012, the REF inversion shows a change from _-5% to _+5% compared to the prior (Figure 1). Wouldn't a change in tropospheric oxidation of 10% over 2 years actually be considered quite substantial, and have major impacts on, for example, the global methane budget?

For comparison with e.g. Montzka et al. (2011), it would be more appropriate to consider the annual mean k.OH anomalies presented in Figure 8. The change in k.OH in 2011, 2012 and 2013 are +5%, +3% and -4% respectively. The reviewer correctly notes that these variations are large with respect to the standard deviation of the interannual variations (2.4%) and it is striking that we find large variations in subsequent years. However, whether this is statistically unexpected is difficult to say: one 2-sigma deviation is expected in a twenty-year timeseries and the distribution of interannual variations over the 20-year time period is not very different from a normal distribution. Also based on Reviewer 1's comments, we have placed more emphasis on the large 2012 anomaly, but we still consider that overall interannual variability is small.

Furthermore, we do think that such OH variations will have a significant impact when applied to the methane budget. This is an aspect we did not investigate thoroughly, because methane was not included in our simulations. As was shown in Rigby et al. (2017) and Turner et al. (2017), interannual variations in OH of a few percent can significantly affect the most likely interpretation of the methane budget, especially when the interannual variations stack up to larger multi-annual variations. This why we recommend inclusion of our derived OH variations in a methane inversion, even if our timeseries of OH is still highly uncertain. We now provide the methane lifetime in Supplement S1, but the effect of lifetime variations on the $CH_4$ budget requires further research.

An additional point: converged solutions in Figure S6 seem to show, in general, more variation than the unconverged "main" results. So, again, it would be important to investigate more fully how sensitive this main conclusion is to the lack of convergence in the main results

The difference between the blue solid line (REF inversion, 20y) and blue dashed line (REF inversion, 10y) in Fig. S6 we consider to be within the error margin of our posterior OH estimate. We draw confidence in the 20-year inversion from the high degree of consistency between the timing of OH anomalies from these two inverse results that have reached different degrees of convergence (excepting a spin-down period).

IAV in posterior k.OH anomalies over 1998-2008 is 2.0% and 2.9% for the 20 and 10 year inversions, respectively. Which of these is better depends on how well we want to fit observations, which brings us back to the difficulty of defining a model error. While the converged state corresponding to our inverse framework will likely have an IAV in OH similar to (or even larger than) that of the ten-year inversion, we can already reproduce atmospheric observations within reasonable bounds with the IAV of the twenty-year REF inversion (e.g. Fig. 6). We have added a paragraph to the discussion on convergence that addresses this issue (Lines 378-386). We still consider the estimate of IAV in OH of <3%, i.e. the number in our abstract, to be consistent with our results, even when we account for the increased amplitude in the ten-year inversions.

3. If emissions are being derived in the main inversions, why was it necessary to "preoptimize" the emissions, assuming constant loss? What happens if you don't do this? If this changes the result substantially, I'd be very concerned, as you're essentially using the observations twice, and, in the first step, you're fixing one of the parameters that you are trying to infer in the second pass. If it doesn't change the results substantially, then wouldn't this step be unnecessary?

We agree that it is not completely statistically sound to pre-optimize emissions with global mean MCF mole fractions, but it was a step necessary for the inversion to converge in a reasonable number of iterations.

Firstly, there is no reliable emission inventory available, especially in the second decade of our inversion, so some arbitrary choice for emissions is necessary. The inverse framework includes relative errors on emissions, because in an inversion with absolute uncertainties we cannot exclude negative emissions, which is problematic when a loss process needs to be separated from emissions. As noted in the reply to Reviewer 1, the relative emission error actually increases from 50% in 2005 to 200% in 2015 linearly, to allow for a wider range of posterior emissions when prior emissions are low (this is now mentioned in lines 120-123 of the manuscript). However, due to the relative error, we still need to define some emissions in later years to allow for a posterior scenario with substantial emissions. We performed a test inversion where we floored emissions at 5 Gg/year: a rather substantial amount. We considered this most fair because it does not include any prior information and allows for many posterior scenarios. In the prior simulation, MCF mole fractions after 2010 (predictably) ended up much too high. The inverse framework compensated with large adjustments in both emissions and OH, because the overestimate was present at all surface sites. We would hope that, eventually, when global mean MCF is captured, the inversion will start matching the small site-to-site gradients and determine that the overestimate is largely driven by too-high emissions.
However, after many iterations we had still not reached that stage, because the large overestimate relative to the small observational error resulted in a very large prior cost that gave the inversion problems. Therefore, we choose to be pragmatic and loosely fit prior emissions to the global MCF growth rate, so that the prior simulation does not drift too far from observations in later years. The prior fit is not even very good (e.g. dashed lines in Fig 4): a "problem" with the pre-optimization that we choose not to correct, because it tests the ability of our inverse system to optimize the global growth rate when the prior simulation is not too poor.

Secondly, we use the global mean mole fractions of MCF now both in the pre-optimization and in the 3D model inversion. However, we argue that the 3D model inversion is driven by additional information contained in the observations: intrahemispheric gradients, vertical gradients (e.g. MLO / KUM), realistic transport, etc. We performed the inversion in the 3D model precisely because we are interested in this latter source of information. Global and hemispheric mean mole fractions of MCF have been explored extensively already in previous box model studies. We considered the double use of global mean mole fractions a price worth paying for easy access to the more interesting information contained in subtle MCF gradients.

**Minor comments:**

L7: ". . . better reproduce. . ." (than what?)

Adjusted: ".. compared to the prior simulations .." (L7)

L18: "the signals are small". This is too vague. Need to specify what quantities are being referred to.
L19: ". . . better match the global MCF observations. . ." (than what?)

Rephrased to: *While the effect of the derived temporal OH variations on MCF mole fractions is small, these variations do result in an improved match with MCF observations relative to an interannually repeating prior for OH. Therefore, we consider the derived variations relevant for studying the budget of e.g. $CH_4$.* (L18-20)

L33 – 35: citation needed.

Citations have been added.

L48: What does the phrase: "but artifact-free sampling has become more difficult" mean?

We wanted to convey that declining MCF mole fractions have led to some issues with contamination during sampling in the NOAA network observations, even if the repeatability of flask pair measurements is not affected. Most notably, for this reason, there are no MCF observations available for SPO during 2015-2016. We have adjusted the text to clarify:

*Through improvements in measurement techniques, measurement quality has mostly kept pace with the atmospheric decline of MCF. However, artifact-free sampling has become more difficult, because small contamination issues that might have been insignificant years ago become substantially more important as the MCF mole fraction has declined.* L49-51

L91 and throughout: phrases like "such as OH" need to specify that it is OH concentration that is being referred to. Perhaps define "[OH]".

We have made adjustments throughout to distinguish between OH, OH concentrations and global mean OH concentrations ($[OH]_{GM}$)

L155: Can you be more specific than saying that model error was "proportional to the 3D spatial gradients". I.e. define what you mean by spatial gradients and let us know if there was a constant of proportionality.

We have changed the phrasing to be more specific and refer to the study that introduced this error set-up for TM5:

*On top of the measurement error, we also included a model representativeness error for each observation. This error is calculated as an absolute average over the mole fraction gradients between the model grid cell that contains an observation and horizontally and vertically adjacent grid cells (Bergamaschi et al., 2005).* L 160-163

From this phrasing we think it is clear that we use no constant of proportionality.

L180: ". . . correlation with the REF inversion" (need to state what the correlation is with respect to)

Adjusted.

L195: I'm not sure what you mean by "would be hard to exclude from a bottom-up perspective". Can you be more explicit?

We want to convey that the absolute magnitude of emissions and emission variations that we derive for these years are very small. While an emission increase from 2012 towards 2013 (derived in the REF inversion) is not expected, the increase is so small that we cannot exclude it based on prior knowledge of emissions during these years. We have adjusted the text:

*Firstly, we note that the small emission totals in later years of around 2 Gg/yr, with interannual variations of 0.2 Gg/yr, would be hard to exclude based on prior knowledge of emissions.* L282-283

Figure 4: There seems to be a consistent increase in the mismatch in the IH gradient. Any idea what causes this? Seems like a potentially interesting feature. Could this point to a sudden increase in emissions?

An increasing relative contribution of emissions to the MCF budget is a potential explanation for this change. We also derive increasing to near-constant MCF emissions after 2013 in the REF inversion (Fig. 3). However, it is difficult to interpret these small differences in IH gradients for MCF. MCF abundance is lowest in the tropics and so the IH gradient, especially when low-latitude sites are included, might not be the most insightful quantity. For example, we quite well capture the Alert to Cape Grim gradient, which could indicate posterior emissions are realistic (Fig. 5). On the other hand, the observed MCF gradient between Alert and Mauna Loa also increases in recent years, relative to global mean MCF, which our simulations do not capture (Fig. 5). This increase is not seen in the Southern Hemispheric Samoa to Cape Grim gradient, which makes an explanation involving MCF emissions more likely.

However, we consider that correct interpretation of changes in the spatial gradients of MCF is too complicated to do without a model. For example, why would a change in emissions in recent years drive an increase in the ALT to MLO gradient, but not in the ALT to CGO gradient? Therefore, we choose not to hypothesize much on the potential drivers of posterior residuals in the manuscript. We only intended to show with Fig. 4 that the inversion performs well also on those quantities (global and hemispheric averages) that were used in previous box model studies.

Figure 5 and throughout: For the reader not familiar with these site codes, it would be useful to clarify where these stations are (i.e. their latitude is particularly important for the discussion of Figure 5).

For the interested reader Table S2 is available. We now also include latitudes next to the site abbreviation in Figure 5 and corresponding supplemental figures.

L374-375: It would seem important to include the results of this test (scaling global OH) in the supplement.

We would agree if the only difference between the inversion with global scaling of OH and the presented inversions was the degrees of freedom we gave to OH. However, we made other changes too after this test inversion (e.g. to the prior emissions) and so a fair presentation of these results would require an extensive and complex explanation. We are afraid that the storyline of the paper would become even more convoluted.

L379: "These results indicate substantial robustness of the derived OH variations". I'm wondering if this can really be stated so strongly, given that the three main inversion show different OH variations?

We have attenuated the statement somewhat to:

*These results indicate that the solution we have derived is robust and consistent with observed gradients in MCF.* L396-397

We do think that while the three twenty-year inversions show different OH variations, all other tests indicate robustness, at least for the timing and approximate amplitude of the derived OH variations.

**References**

1. Wennberg, Paul O., et al. "Recent changes in the air-sea gas exchange of methyl chloroform." *Geophysical research letters* 31.16 (2004).

2. Montzka, Stephen A., et al. "Small interannual variability of global atmospheric hydroxyl." *Science* 331.6013 (2011): 67-69.

3. Turner, Alexander J., et al. "Ambiguity in the causes for decadal trends in atmospheric methane and hydroxyl." *Proceedings of the National Academy of Sciences* 114.21 (2017): 5367-5372.

4. Rigby, Matthew, et al. "Role of atmospheric oxidation in recent methane growth." *Proceedings of the National Academy of Sciences* 114.21 (2017): 5373-5377.

5. Naus, Stijn, et al. "Constraints and biases in a tropospheric two-box model of OH." *Atmospheric Chemistry and Physics* 19.1 (2019): 407-424.

---

## Author Comment (AC3) · 8 Dec 2020

L349: What are the criteria for stopping the iteration? Are the 3 inversion experiments reach a similar value of cost function J (and the gradient of the cost function) in the last iteration?

The inversions were stopped when additional iterations did not result in further modification of the solution, e.g. the OH variations. Because of the reasons outlined in this section, this did not necessarily mean that the cost function gradient was reduced by a factor that we pre-defined (typically 1000). Rather, we consider the stagnation of the inversion indicative of convergence problems.

The REF and TM5OH 20-year inversions converge to a similar value for the cost function, while the POP inversion ends up somewhat higher. However, this is just one number: we consider an analysis of the error statistics of the solution more insightful, such as in Figure 6.

The norm of the cost function gradient is typically reduced by a factor ~100 in all three inversions, but this value strongly oscillates between subsequent iterations.

L352: Why further convergence will result in less realistic OH variations?

The 10-year inversions find large adjustments in the latitudinal distribution of OH that fall significantly outside the spread of a wide range of literature estimates (see the new Figure S8). Therefore, we did not consider it worth the computational cost to continue the 20-year inversions only to end up closer to the 10-year inversions, especially since we consider some aspects of the 10-year inverse solution physically unrealistic.

L372: "Firstly...we generally identified similar tendencies in each." Figure 1 has shown the variations of OH estimated by three inversions are quite different.

Our argument is that this difference comes from convergence issues and not from a difference in the converged solution. The evidence for this is that the 10-year inversions do convergence to the same OH variations, which makes it very likely that the 20-year inversions would also, given infinite computational time. Moreover, in Figure 1 it can also be seen that there are similarities between the three inversions in the timing and sign of derived OH variations, only these OH variations are most pronounced in the REF inversion. More precisely, a positive OH anomaly in the REF solution never coincides with a negative anomaly in the TM5OH and POP solutions, and vice versa.

L374-379: Here the manuscript tried to prove the robustness of the OH interannual by an additional inversion and a forward simulation. But the details of the two experiments and the results are not given. I suggest include some details in the supplements. E.g. how the one global scaling factors compare with the REF, POP, and TM5OH? Is the forward simulation use the prior or optimized MCF emissions? I cannot understand the logic here, can you clarify why the two experiments can indicate the robustness of the derived OH variations?

The point we want to make here is that the interannual variations in global mean [OH] (or oxidation) give an improvement relative to interannually repeating OH that is independent of the large latitudinal OH adjustments we find. I.e., even if the latitudinal biases were less strong (for example if we adjust the ocean flux) we would likely still find the same OH variations.

The simplest way that we have shown this is by only including the global OH variations (as in Figure 1) in a forward simulation, and not the latitudinal OH adjustments. This forward simulation reproduces the observations better than a forward simulation with no interannual OH variations. We find this result even if we use prior MCF emissions in both simulations (although, obviously, if we also include optimized MCF emission variations, the result improves further).

The second test that we performed in an early stage of the study is to only optimize one global scaling factor for OH. In this case, we still derived similar OH variations. We choose not to present the inversions with one OH scaling factor, because in these test inversions several other input variables were different from the inversions presented in the main manuscript (e.g. the prior emissions). Therefore, a fair presentation of these results would require an extensive and complex explanation. We are afraid that the storyline of the paper would become even more convoluted.

We only mention these two experiments to underline the (minor) point that the derived global OH variations are independent of the latitudinal OH adjustments. The general robustness of the derived OH variations is best shown by the ten-year inversions, which we do discuss extensively.

---

## Author Response (AR1)

A point-by-point response to the reviewer comments is included in the response to reviewer's. This document marks the changes between the previous and the new manuscript version.

[revised manuscript text omitted]

---

## Referee Report (RR1)

The authors discussed the comments in the response letter. However, only a few of the comments are addressed in the manuscript. I recommend the authors go through all the comments again and at least include most of the comments in the revised manuscript. In addition, I still have some comments that are not well discussed in the response letter. I think at least a major revision is needed.

Other specific comments:

1 Before inversion, the prior emissions were pre-optimized to fit the global mean MCF mole fractions. The authors argue that the pre-optimized MCF emissions can reduce computational costs. But the inversion as shown by equation(1) is to estimate the emissions and OH by combining the information from both bottom-up estimated prior emission inventories and the observations, as well as their errors. The pre-optimize erase the information of prior emission inventories, and only keep the information of observations. I don't think this is the right way to do an inversion. Besides, the question is not answered:" Will the pre-optimization reduce the OH variation estimated by the inversions since the MCF emissions already fit the observations?" In addition, from the author's response to Fig.4, I feel the pre-optimization is somewhat arbitrary, which makes the inversion lost the prior information.

2 The authors explain why the small correlation coefficient between OH and MCF variations can reflect the OH and MCF are independently derived. But I think a better indicator should be comparing optimized minus prior MCF emissions and optimized minus prior OH. We can see that in Fig.3, the MCF emissions estimated by REF inversion is much higher than prior around 2013, which is corresponding to the large positive OH anomaly around 2012-2013. This may indicate that the inversion system cannot separate the OH and MCF variations.

3 For the convergence problems. From the author's discussion, the 10-year inversions can reach convergence since they require less time per iteration. One problem is that the 10 years inversions are for 1998-2008 when the MCF emissions are higher than 2009-2018 and the corresponding errors are much lower than 2009-2018. If the 10 years inversions focus on 2009-2018, it will be hard to say if the 10-year inversion coverage to similar OH variations since the uncertainties in MCF emissions (reach 200%) are much larger OH during 2009-2018. So I don't think the 10-year inversion for 1998-2008 can prove the robustness of the 20-year inversions, as the author mentioned when discussed my last comment.

4 "L352: Why further convergence will result in less realistic OH variations?"
The authors answer this question by adding Figure S8 (but show nothing in the manuscript) which showed that the inversions are overfitting. Is this because the inversions use too small observational error?

5 "L372: "Firstly...we generally identified similar tendencies in each." Figure 1 has

shown the variations of OH estimated by three inversions are quite different."

The authors answer this by showing the 10 years inversions are similar. But as aforementioned, the three 10-year inversions are similar may not prove the inversions for 2009-2018 can also reach similar results. Here the only thing we see in the main text is that the three 20-year inversions are quite different.

6 "L374-379: Here the manuscript tried to prove the robustness of the OH interannual by an additional inversion and a forward simulation. But the details of the two experiments and the results are not given. I suggest include some details in the supplements. E.g. how the one global scaling factors compare with the REF, POP, and TM5OH? Is the forward simulation use the prior or optimized MCF emissions? I cannot understand the logic here, can you clarify why the two experiments can indicate the robustness of the derived OH variations?"

It is still unclear how the authors conduct the two experiments. I think every model experiment established should be introduced in the manuscript or supplements. Since the two model experiments are not shown clearly, the role of the two experiments is certainly unclear.

---

## Author Response (AR2)

Dear authors, your manuscript will have to undergo a second round of review. But before this is initiiated, please complete the answer t the first review first. I could not find the answer to the last of remarks of reviewer #1 on the page C4 of RC1.

Also I believe, that you missed part of the first question of reviewer #2.

In my understanding, the reviewers raised critical points which you did not answer. If I am wrong, please point me to the answers to this questions.

In the process of checking your answers to the referees, please check again, if you completed the review.

Kind regards
Mathias Palm

Thank you for pointing us to some of the comments that we had missed before.

Firstly, we have now addressed the comments on page C4 of Reviewer #1 a Part 2 of our response, which we have posted in the discussion. The comments we had missed did not require additional changes to the manuscript, so that we have uploaded the same version again.

Secondly, you mention we did not fully address reviewer #2's opening statement. While we do not explicitly address each minor aspect, I think we do address the main concerns as best as we can by further elaborating on our choices. If you feel that we still need to address a specific aspect of the reviewers' comments, please let us know.

The bottom line of our response is that we cannot change the design of the study as this point. Instead, we have added new material to support our choices in the revisions. Here we honestly report convergence problems. If these convergence problems are a fundamental reason to not publish our results, then there is little we can do at this point. However, we think this would be a mistake, because there is still much value in the results we present. On top of that, our experience should be known to anyone that will attempt to perform a similarly meticulous inversion of MCF.

---

## Author Response (AR3)

Review #1

The authors discussed the comments in the response letter. However, only a few of the comments are addressed in the manuscript. I recommend the authors go through all the comments again and at least include most of the comments in the revised manuscript. In addition, I still have some comments that are not well discussed in the response letter. I think at least a major revision is needed.

We thank the reviewer for considering the revised manuscript for publication. We understand that the reviewer considers some of their comments sufficiently covered by our response letter and would like to see more significant changes in the manuscript to complement our response. In addition, some of the comments were not addressed sufficiently in the direct response, which are outlined below.

Other specific comments:

1 Before inversion, the prior emissions were pre-optimized to fit the global mean MCF mole fractions. The authors argue that the pre-optimized MCF emissions can reduce computational costs. But the inversion as shown by equation(1) is to estimate the emissions and OH by combining the information from both bottom-up estimated prior emission inventories and the observations, as well as their errors. The pre-optimize erase the information of prior emission inventories, and only keep the information of observations. I don't think this is the right way to do an inversion. Besides, the question is not answered:" Will the pre-optimization reduce the OH variation estimated by the inversions since the MCF emissions already fit the observations?" In addition, from the author's response to Fig.4, I feel the pre-optimization is somewhat arbitrary, which makes the inversion lost the prior information.

[Figure]

**Fig. R1**: *The MCF emissions that result from fitting to global mean MCF mole fractions (blue, dashed), and the 3-year smoothed emissions that are used in our inversions.*

Of course, this is an important and valid remark that we gave considerable thoughts. Firstly, we note that after the pre-fitting to annual, global mean MCF mole fractions, prior emissions are smoothed with a 3-year moving average: precisely to avoid pre-fitting interannual variability of MCF too heavily, and because there is no clear bottom-up explanation for strong interannual variations in MCF emissions. Fig. R1 shows that this smoothing helps to remove large interannual variations in emissions, though slower, multi-annual variations remain. The reviewers' remark made us realize that we had not mentioned the emission smoothing before in our response nor in the manuscript but we have now added it in L120-129.

We emphasize that in later years especially, no prior information on MCF emissions is available, and any emission prior therefore requires arbitrary assumptions. Moreover, the global mean MCF mole fraction is only a small part of the observational information that the inverse system leverages to obtain OH and emission variations. Observed spatio-temporal gradients in MCF and transport variations are key to the novelty of our work, compared to previous box model inversions, and this information is completely unused in the prior. We argue that by pre-optimizing emissions, the inversion more quickly starts with optimizing the spatial gradients that are most interesting and bring relevant information to the OH distribution. We now additionally describe and motivate this choice in the manuscript (L120-129).

2 The authors explain why the small correlation coefficient between OH and MCF variations can reflect the OH and MCF are independently derived. But I think a better indicator should be comparing optimized minus prior MCF emissions and optimized minus prior OH. We can see that in Fig.3, the MCF emissions estimated by REF inversion is much higher than prior around 2013, which is corresponding to the large positive OH anomaly around 2012-2013. This may indicate that the inversion system cannot separate the OH and MCF variations.

The reviewer's suggestion is exactly what the correlation coefficient quantifies: it is the correlation between optimized interannual OH variations and interannual variations relative to prior emissions (i.e. as a percentage of the prior emissions). In other words, the inversion derives emission and OH variations that are not correlated. We have now clarified this in the text (L221-223).

Furthermore, we note that the peak in MCF emissions is highest in 2013-14, while the OH anomaly peaks in 2012 and quickly declines into 2013-14. Therefore, while this large anomaly in both quantities is remarkable, the two are not directly coincident in time. Moreover, a positive anomaly in both quantities is undesirable from a cost-function perspective, since positive anomalies in both will cancel out on a global scale, but they will increase the background cost. Therefore, based on our understanding of the inverse system, we consider these variations driven by observational information from the spatial gradients in MCF that require compensating variations in these two state parameters that have distinctly different spatial imprints on MCF.

3 For the convergence problems. From the author's discussion, the 10-year inversions can reach convergence since they require less time per iteration. One problem is that the 10 years inversions are for 1998-2008 when the MCF emissions are higher than 2009-2018 and the corresponding errors are much lower than 2009-2018. If the 10 years inversions focus on 2009-2018, it will be hard to say if the 10-year inversion coverage to similar OH variations since the uncertainties in MCF emissions (reach 200%) are much larger OH during 2009-2018. So I don't think the 10-year inversion for 1998-2008 can prove the robustness of the 20-year inversions, as the author mentioned when discussed my last comment.

This is an excellent point that was insufficiently covered in the manuscript. The spatial signatures of OH and MCF emissions remain distinctly different also in later years, which is part of what drives the system's skill in deriving both OH and emission variations. However, the increased emission uncertainty relative to the atmospheric burden of MCF will likely impact the robustness of the system.

Between 2008 and 2013, the increase in relative uncertainty of emissions is compensated for by a sharp decline in emissions over this period. However, after 2013, prior emissions stabilize and the absolute prior error on emissions becomes significantly larger than the absolute prior error on OH (which can be approximated by assuming that 20% of the MCF burden is removed by OH each year). We now explicitly mention this in the manuscript as a point of caution for the interpretation of derived variations, especially after 2013.

We now emphasize this point in several places in the manuscript (L228-232; L412-417; L503-505)

4 "L352: Why further convergence will result in less realistic OH variations?" The authors answer this question by adding Figure S8 (but show nothing in the manuscript) which showed that the inversions are overfitting. Is this because the inversions use too small observational error?

We do make the case that perhaps the error we use on observations is overly optimistic (L264-267). However, this was motivated by the inability of our coarse-resolution simulations to capture short-term variations in MCF, not by the systematic intrahemispheric biases. We consider these systematic biases to fall outside the bounds of realistic observational and model errors, and therefore propose a changed MCF ocean flux as a more likely explanation. In other words, we consider this "overfitting" a result of including insufficient degrees of freedom in the state, rather than of too-small errors on observations.

We have slightly expanded the reference to Fig. S8 in the new manuscript (L215-216; L331-332).

5 "L372: "Firstly...we generally identified similar tendencies in each." Figure 1 has shown the variations of OH estimated by three inversions are quite different." The authors answer this by

showing the 10 years inversions are similar. But as aforementioned, the three 10-year inversions are similar may not prove the inversions for 2009-2018 can also reach similar results. Here the only thing we see in the main text is that the three 20-year inversions are quite different.

This phrasing was indeed too strong, and we have rewritten this paragraph (L408-417). We mainly view it as important that the derived OH variations are never opposite between inversion set-ups (hence, have the same "tendencies"). Opposite variations would invalidate the premise that the different inversions are moving in the same direction.

6 "L374-379: Here the manuscript tried to prove the robustness of the OH interannual by an additional inversion and a forward simulation. But the details of the two experiments and the results are not given. I suggest include some details in the supplements. E.g. how the one global scaling factors compare with the REF, POP, and TM5OH? Is the forward simulation use the prior or optimized MCF emissions? I cannot understand the logic here, can you clarify why the two experiments can indicate the robustness of the derived OH variations?"

It is still unclear how the authors conduct the two experiments. I think every model experiment established should be introduced in the manuscript or supplements. Since the two model experiments are not shown clearly, the role of the two experiments is certainly unclear.

Our main concern that we aimed to address with these two experiments was that we find only small variations in global mean oxidation (few percent) compared to the systematic spatial adjustments to the OH distribution (tens of percent). Therefore, we doubted how vital these global-scale variations were to the solution and the derived cost-function reduction. We consider this result not important enough for the already complex storyline of our study to extensively discuss them, which is why we chose to remove mention of them.

As noted by the reviewer, it remains difficult to explicitly prove robustness of the REF solution over the 2008-2018 period, and so we now clearly mark this distinction in the manuscript, as noted in point 3. Most notably, we have rewritten the first paragraph of the discussion (L408-417).

Motivated by the new and old reviews, we have made additional adjustments to the manuscript that better highlight the important qualities of our work. For example, we provide a more nuanced perspective in L408-417, and we provide suggestions for improvements in L448-462.

Review #2

I thank the authors for giving the reviewer comments careful consideration.

The authors do not propose too many major changes to the paper as a result of these comments. They acknowledge that there are some limitations to their study (e.g. lack of convergence), but make the case that it is too late to change the experimental design, and argue that there is enough useful information in the paper to warrant publication. I am inclined to agree with this assessment. However, I suggest that the limitations should be spelled out more clearly in the abstract. I suggest adding some text along the following lines (it doesn't have to be exactly this, but I think these caveats need to be noted):

We thank the reviewer for making such helpful and specific suggestions that help improve our manuscript.

- Title: I'm still quite concerned about the lack of convergence. Therefore, I wonder if it's more accurate to remove "inversion" from the title. Perhaps something like: "An investigation of top-down constraints on atmospheric oxidative capacity using methyl chloroform and a global 3D model"

We understand the reviewer's concern. However, the timeseries of OH variations we derive, even if not fully converged in the 20-year inversions, incorporates the spatiotemporal gradients of methyl chloroform observations in a way that only a 3D model inversion can. Recently, the study of Patra et al. (2020) was published, which is also well-described by the suggested title, but did not include a 3D model inversion. The title doesn't suggest any definite conclusions regarding OH variability or trends, which would not be supported by our work. Therefore, we strongly prefer to have 3D model inversion in our title.

If the reviewer still considers this title too bold, we are willing to change it.

- At the end of the first paragraph, or start of the second paragraph, I suggest noting the lack of convergence (e.g. "While our main solutions did not fully converge, they suggest interannual variations in the global oxidative capacity...")

Now noted in L6.

- When talking about the inter-annual variability, I suggest being explicit that the standard deviation of the derived OH concentration was small (< 3% per year) over the 20 year period, but that substantial fluctuations (~±10 %) were derived between certain years.

We have now added a comment that explicitly addresses the rapid change from 2010 to 2012 (8% in annual mean values), in the abstract (L10-12) as well as in the conclusion (495-496). However, as noted in our previous response, we find that the twenty-year OH variability is reasonably well described using a standard deviation of < 3%.

- At the start of the third paragraph of the abstract, it needs to be made clear what the "adjustments" are with respect to (i.e. "compared to a widely used prior OH distribution…", or perhaps even cite the relevant dataset).

Now noted in L14. We do not cite the relevant dataset, because similar adjustments are found for the two different OH distributions.

- At the start of the fourth paragraph of the abstract, clarify (perhaps in parentheses) broadly what the added value of the 3D model is.

We have added clarification in L20-21.

- Final sentence of the abstract: Given the lack of convergence, I think it is too strong to suggest that these particular results be used in studies of the CH4 budget (which is implied), and therefore suggest that this be softened to "Therefore, we consider that variations in OH derived from MCF inversions with 3D models can add value to studies of the budget of e.g. CH4"

We have softened this statement as suggested (L23-24).

In the supplement, line 35: Which section do you mean S2?

Section 2.1.1, it is filled in now.

**Reference**

Patra, P. K., et al. "Methyl Chloroform continues to constrain the hydroxyl (OH) variability in the troposphere." Journal of Geophysical Research: Atmospheres: *e2020JD033862*.